# Automata Learning and Identification of the Support of Language Models

**Satwik Bhattamishra**[1]* **Michael Hahn**[2] **Varun Kanade**[1]

[1]University of Oxford [2]Saarland University

## Abstract

We study the learnability of languages in the *Next Symbol Prediction* (NSP) setting, where a learner receives only positive examples from a language together with, for every prefix, (i) whether the prefix itself is in the language and (ii) which next symbols can lead to an accepting string. This setting has been used in prior work to empirically analyze neural sequence models, and additionally, we observe that efficient algorithms for the NSP setting can be used to learn the (truncated) support of language models. We first show that the class of DFAs with at most $n$ states is identifiable from positive examples augmented with these NSP labels. Nevertheless, even with this richer supervision, we show that PAC-learning DFAs remains computationally hard, and exact identification using only membership queries cannot be achieved in polynomial time. We then present $L^\star_{\mathrm{nsp}}$, an extension of Angluin's $L^\star$ algorithm, and show that DFAs can be PAC-learned efficiently using a language-model–based teacher that answers membership queries and generates valid strings conditioned on prefix prompts. Finally, we conduct a comprehensive experimental evaluation on 11 regular languages of varying complexity. Using $L^\star_{\mathrm{nsp}}$, we extract DFAs from Transformer-based language models trained on regular languages to evaluate the algorithm's effectiveness and identify erroneous examples.

## 1 Introduction

Language models (LMs) are now deployed across text, vision, and bioinformatics; yet their internal computation and potential outputs they generate remain difficult to interpret. This motivates a basic question: given black-box access to a model, can we extract a compact, interpretable formal object, such as a deterministic finite automaton (DFA), that accepts (approximately) the same strings as those that lie in the model's generative support? We develop a formal framework for this problem and study its learnability in a setting that closely mirrors how LMs are typically used in practice.

We formalize and study the problem of learnability of languages in the *Next Symbol Prediction* (NSP) setting. Here a learner receives only *positive* strings from a target language, together with rich supervision for every prefix: a membership bit indicating whether the prefix itself is in the language and a vector of "continuation" bits indicating which next symbols admit some accepting continuation. A prediction is correct only if the hypothesis matches *all* membership and continuation labels at *every* prefix of the example.

This setup has a natural interpretation in the context of generative models: when decoding with top-$p$ (Holtzman et al., 2019), top-$k$, or min-$p$ (Minh et al., 2025) sampling, the per-prefix continuation set is precisely the set of admissible next tokens, and termination corresponds to allowing the end-of-sequence token. In particular, positive-only NSP supervision is natural to obtain from black-box models and avoids inventing an artificial distribution over negative strings; at the same time, the requirement for correct predictions at every prefix makes the task challenging. Fig. 1 illustrates NSP labels on a Dyck example and how admissible-next-token sets can be read from an LM.

Additionally, while NSP has been widely used to *evaluate* sequence models on formal-language benchmarks (see e.g. Gers & Schmidhuber (2001); Suzgun et al. (2019); Ebrahimi et al. (2020);

---

*Corresponding author. Contact: satwik.bmishra@cs.ox.ac.uk

Bhattamishra et al. (2020a) and references therein), the learnability of languages under NSP and its relationship to conventional binary classification has not been established.

**Our contributions.** We investigate the question of learnability in the NSP setting within the computational learning theory framework by studying computational complexity and oracle requirements. We also conduct a systematic empirical evaluation with several regular languages, using Transformer-based language models as teachers.

**(i) Identifiability and hardness.** We give a PAC-style formulation of learning using NSP labels and show that positive examples augmented with NSP labels are information-theoretically sufficient to identify minimal DFAs. Concretely, distinct minimal DFAs always disagree on the NSP labeling of some positive string, yielding finite teaching sets and a well-defined equivalence oracle in the NSP model. At the same time, we prove that NSP supervision does *not* remove the key *computational barriers* for learning regular languages. The key technical argument is a construction that renders all but one continuation label uninformative, allowing a reduction to well-known hardness results (Kearns & Valiant, 1994). Thus, even with the richer labels, efficient (improper) learning remains computationally intractable (under standard cryptographic assumptions). We further show that identification with membership queries alone cannot be achieved in polynomial time for certain natural DFA families, even when each query returns all NSP labels. Together, these results suggest that while NSP labels offer some benefit, they do not, in general, circumvent computational hardness.

**(ii) Learning with a language-model teacher.** Motivated by the hardness results, we study a more powerful, though still practically motivated, learning framework based on prefix-conditional generation queries. These can easily be simulated using black-box access to an LM. In addition to membership queries, the learner can issue *generative* queries that return positively labeled NSP strings conditioned on a *prefix* prompt. We extend Angluin's $L^\star$ algorithm (Angluin, 1987) to design a new algorithm we denote $L^\star_{\mathrm{nsp}}$ that uses membership and generative queries to construct a DFA consistent with the observed NSP labels. The guarantee is distribution-specific: $L^\star_{\mathrm{nsp}}$ PAC-learns with respect to the distribution induced by the teacher language model. This perspective is aligned with our goal of identifying the model's (truncated) support and is particularly appealing in the context of generative models, where the target distribution over negative strings is either undefined or arbitrary. Conditional generation is both a natural capability of modern LMs and turns out to be a powerful query primitive for efficient learning in the NSP framework.

**(iii) Empirical evaluation and analysis.** We apply the $L^\star_{\mathrm{nsp}}$ algorithm to extract DFAs from Transformer teachers trained on eleven regular languages of varying complexity, including the Tomita grammars, Parity, and bounded Dyck languages. We study how NSP accuracy, the number of extracted states, and runtime scale with the number of positive training strings. Across tasks, a modest amount of positive data with NSP labels typically suffices to recover the target automata or their teacher-support counterparts. When the teacher is imperfect (e.g., for Parity, Tomita-5), the extracted DFA reveals systematic errors by identifying strings in the symmetric difference between the target and the teacher's support. Ablations further indicate that the continuation labels are heavily used on languages with transitions to dead states (e.g., bounded Dyck), leading to improved sample complexity over binary labels alone. These experiments underscore a practical point: while NSP labels cannot break worst-case computational barriers, they are easy to obtain from modern sequence models and can be leveraged effectively in practice.

## 1.1 RELATED WORK

**Learnability of DFAs.** Classical results show that inferring DFAs from labeled examples is computationally hard— finding the minimum consistent DFA is NP-hard (Gold, 1978; Angluin, 1978; Pitt & Warmuth, 1993), and even (improper) PAC learning is intractable under cryptographic assumptions (Kearns & Valiant, 1994). With queries and counterexamples, Angluin (1987) showed that regular languages are learnable in polynomial time. Under structural assumptions, several works have studied PAC learnability for probabilistic DFAs (Clark & Thollard, 2004; Palmer & Goldberg, 2007). The area remains active with recent theory on counterexample handling and lower bounds in the MAT framework (Vaandrager et al., 2022; Kruger et al., 2023). We build on this line of research by characterizing the learnability of DFAs in a new setting relevant to learning the support of language models, and by situating prior empirical analyses of neural nets in the appropriate context.

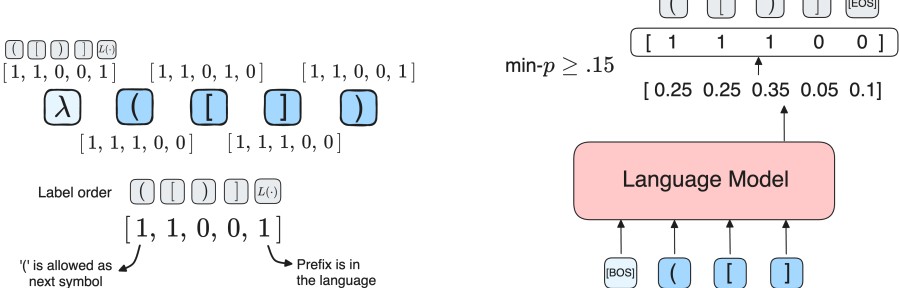

Figure 1: *Left:* An example of a string with NSP labels from the Dyck-2 language. The language consists of well-balanced parentheses with two types of brackets. *Right:* An illustration of how NSP labels can be obtained from a language model. See Section 2 for details.

**Automata extraction from neural models** was introduced by Giles et al. (1992); Omlin & Giles (1996) and has been an active field (Wang et al., 2018; Muškardin et al., 2022). Notably, Weiss et al. (2018) developed a method for white-box extraction based on L⋆, and several subsequent works have focused on weighted automata/PDFAs (Weiss et al., 2019; Wei et al., 2024; Eyraud & Ayache, 2024). Recent work (Zhang et al., 2024) has also explored L⋆-like methods for Transformer-based classifiers. Mayr et al. (2023) learn a PDFA abstraction of an LM via a distributional congruence. Our focus is on support identification, where we characterize learnability and conduct empirical analysis based on a new provable extension of the L⋆ algorithm.

## 2 PROBLEM DEFINITION

**Notation.** A deterministic finite automaton (DFA) is a tuple $A = (Q, \Sigma, \delta, q_0, F_A)$ with finite state set $Q$, alphabet $\Sigma$, transition function $\delta : Q \times \Sigma \to Q$, start state $q_0$, and a subset of accepting states $F_A \subseteq Q$. The language of $A$ is $L_A \subseteq \Sigma^*$; write $A(x) = L_A(x) \in \{0, 1\}$. Fix an order $\Sigma = \{\sigma_1, \ldots, \sigma_{|\Sigma|}\}$. For $x = w_1 \cdots w_N$, let the length-$n$ prefix be $x_{:n} := w_1 \cdots w_n$ for $0 \le n \le N$. We use $q_{\text{dead}}$ for a dead state with $\delta(q_{\text{dead}}, \sigma) = q_{\text{dead}}$ for all $\sigma \in \Sigma$ (unique, if present, in a minimal DFA). We use $\text{DFA}_n$ to denote DFAs with at most $n$ states.

### 2.1 NEXT SYMBOL PREDICTION (NSP) SETTING

For a language $L \subseteq \Sigma^*$ and any $x \in \Sigma^*, \sigma \in \Sigma$, define the *continuation bit*

$$\varphi_L(x, \sigma) := \mathbb{I}\big[\exists s \in \Sigma^* \text{ s.t. } x \cdot \sigma \cdot s \in L\big].$$

If $L$ is regular with minimal DFA $A$, then with $q = \delta(q_0, x)$ we have $\varphi_A(x, \sigma) = 0$ if and only if $\delta(q, \sigma) = q_{\text{dead}}$. The continuation vector at $q$ is defined as $\varphi_A(q) = [\varphi_A(q, \sigma_1), \ldots, \varphi_A(q, \sigma_{|\Sigma|})] \in \{0, 1\}^{|\Sigma|}$. For strings $x \in \Sigma^*$, write $\varphi_A(x) := \varphi_A(\delta(q_0, x))$ and $\varphi_A(x, \sigma) := \varphi_A(\delta(q_0, x), \sigma)$.

A positive NSP–labeled example is a string $x = w_1 \cdots w_N \in L$ together with, for every prefix $x_{:n}$ $(0 \le n \le N)$, its membership $L(x_{:n})$ and all continuation bits $(\varphi(x_{:n}, \sigma))_{\sigma \in \Sigma}$. We collect these as

$$f_L(x) := \Big((\varphi(x_{:n}, \sigma_i))_{i=1}^{|\Sigma|}, L(x_{:n})\Big)_{n=0}^{N} \in \{0, 1\}^{(|\Sigma|+1)(N+1)}.$$

We will use the term NSP labels to refer to such labels (See Fig. 1, left for an illustration). We instantiate predictors via automata. For a DFA $A$ define $f_A(x) := \big((\varphi(x_{:n}, \sigma_i))_{i=1}^{|\Sigma|}, L_A(x_{:n})\big)_{n=0}^{|x|}$. Let $f_{A^\star}$ denote the target NSP labeling function. The per-example loss is the $0/1$ sup-norm mismatch $\text{err}(f_A(x), f_{A^\star}(x)) := \|f_A(x) - f_{A^\star}(x)\|_\infty$, i.e., it equals 1 iff *any* membership/continuation label for any prefix is wrong; the NSP loss on $D$ (supported on strings in $L_A^\star$) is $\mathcal{L}_{\text{NSP}}(f_A; f_{A^\star}, D) := \mathbb{E}_{x \sim D}[\text{err}(f_A(x), f_{A^\star}(x))]$. Because all $(|\Sigma| + 1)(|x| + 1)$ labels must be simultaneously correct, NSP is stringent in the sense that a naive random guesser has a near-zero chance of zero error on a typical example (contrast with $\approx 50\%$ in binary classification).

## 2.2 LEARNING THE TRUNCATED SUPPORT OF LANGUAGE MODELS

Let $\Sigma = \mathcal{V} \cup \{[\text{EOS}]\}$ and let a language model LM define next-token probabilities $p_{\text{LM}}(\cdot \mid y)$ on $\Sigma$ for each prefix $y \in \Sigma^*$. A sampling/truncation rule $\mathcal{T}$ (e.g., top-$p$, top-$k$, per-step min-$p$) maps $y$ to the admissible next-symbol set $\mathcal{C}_{\mathcal{T}}(y) \subseteq \Sigma$, with $[\text{EOS}] \in \mathcal{C}_{\mathcal{T}}(y)$ exactly when $y$ may terminate (see Fig. 1, right for an illustration). The $\mathcal{T}$-*truncated support* of LM is the set of all strings that can be generated by running LM under $\mathcal{T}$. Formally,

$$L_{\text{LM}}^{\mathcal{T}} := \Big\{ x = w_1 \cdots w_N : \forall 0 \leq n < N, \ w_{n+1} \in \mathcal{C}_{\mathcal{T}}(w_1 \cdots w_n) \text{ and } [\text{EOS}] \in \mathcal{C}_{\mathcal{T}}(x) \Big\}.$$

The NSP labelling oracle induced by the language model LM and truncation strategy $\mathcal{T}$ is then,

$$L_{\text{LM}}^{\mathcal{T}}(y) := \mathbb{I}[[\text{EOS}] \in \mathcal{C}_{\mathcal{T}}(y)], \qquad \varphi^{\mathcal{T}}(y, \sigma) := \mathbb{I}[\sigma \in \mathcal{C}_{\mathcal{T}}(y)] \ (\sigma \in \Sigma),$$

and we set $f_{\text{LM}}^{\mathcal{T}}(x) := \big( (\varphi^{\mathcal{T}}(x_{:n}, \sigma_i))_{i=1}^{|\Sigma|}, \ L_{\text{LM}}^{\mathcal{T}}(x_{:n}) \big)_{n=0}^{|x|}$. If generation under $\mathcal{T}$ terminates almost surely, then for any admissible step there exists a finite accepting continuation, so $\varphi^{\mathcal{T}}$ coincides with the global NSP semantics above.

**From NSP learning to learning support.** Let $\mathcal{D}_{\text{LM}}^{\mathcal{T}}$ be the distribution of strings generated by LM under $\mathcal{T}$. Any PAC learner that, from NSP-labeled positive examples $(x, f^\star(x))$, outputs $\hat{f}$ with $\mathcal{L}_{\text{NSP}}(\hat{f}; f^\star, \mathcal{D}) \leq \epsilon$ immediately yields, by instantiating $f^\star = f_{\text{LM}}^{\mathcal{T}}$ and $\mathcal{D} = \mathcal{D}_{\text{LM}}^{\mathcal{T}}$, a procedure that learns the $\mathcal{T}$-truncated support of LM with NSP error at most $\epsilon$.

**Oracle simulation.** With black-box access to LM and rule $\mathcal{T}$, one can simulate the typical example oracle $\text{EX}(f_{\text{LM}}^{\mathcal{T}}; \mathcal{D}_{\text{LM}}^{\mathcal{T}})$ by sampling $x \sim \mathcal{D}_{\text{LM}}^{\mathcal{T}}$ and returning $(x, f_{\text{LM}}^{\mathcal{T}}(x))$. Membership queries can be computed by checking if a string takes an admissible path based on the truncation strategy $\mathcal{T}$ and if $[\text{EOS}]$ is permissible at the last step.

## 3 IDENTIFIABILITY AND EQUIVALENCE IN THE NSP SETTING

Before studying efficient learnability, it is first necessary to establish whether NSP labels are *necessary* and *sufficient*, in an information-theoretic sense, to identify a target language from positive examples alone. By *unique identification from positive* NSP *data* for a target DFA $A^\star$ with $L_{A^\star} \neq \emptyset$, we mean that there exists a finite set $S \subseteq L_{A^\star}$ such that

$$\forall A \in \text{DFA}_n, \qquad \Big( \forall x \in S, \ f_A(x) = f_{A^\star}(x) \Big) \implies A \equiv A^\star. \tag{1}$$

Any such $S$ will be called a (positive) NSP *teaching set* for $A^\star$.

Note that positive strings alone without additional labels are not sufficient for such identifiability: over $\Sigma = \{0, 1\}$, let $L_A = \Sigma^*$ and $L_{A^\star} = 1^*$. Every positive example $x \in 1^*$ is accepted by both, so no positive counterexample exists. The same obstruction persists even if, in addition, the oracle reveals the *membership of each prefix*: for any $x \in 1^*$ and any prefix $y$ of $x$ we have $L_A(y) = L_{A^\star}(y) = 1$, so positive examples with prefix-membership labels still cannot distinguish $A$ from $A^\star$. The key property of the NSP labels is that the continuation bits convey information about strings *not* in the language: $\varphi(y, \sigma) = 0$ certifies that no extension of $y\sigma$ is accepted. This additional information suffices to separate distinct minimal DFAs using positive examples only.

**Proposition 3.1.** *Let* $A \neq A^\star$ *be minimal DFAs with* $L_{A^\star} \neq \emptyset$. *Then there exists* $x \in L_{A^\star}$ *such that* $f_A(x) \neq f_{A^\star}(x)$. *Equivalently, the oracle* $\text{EQ}(A; A^\star)$ *is well-defined: it either returns "equivalent" or a positive counterexample* $(x, f_{A^\star}(x))$.

The proof is in Appendix F.

**Consequences.** Proposition 3.1 has two immediate consequences. First, finite teaching sets exist: for each $A \in \text{DFA}_n \setminus \{A^\star\}$, choose a witness $x_A \in L_{A^\star}$ with $f_A(x_A) \neq f_{A^\star}(x_A)$ guaranteed by the proposition, and set $S := \{x_A : A \in \text{DFA}_n \setminus \{A^\star\}\} \subseteq L_{A^\star}$. Then $S$ is a positive NSP teaching set for $A^\star$ in the sense of (1). Second, since it also implies that equivalence query oracle (cf. App. F) is well-defined in the NSP setting, which is crucial for exact learning to be feasible.

## 4 Hardness of Learning

We study efficient PAC learnability in the NSP setting. An algorithm $\mathcal{A}$ is an efficient PAC learner for a class $\mathcal{F}$ if for every $f \in \mathcal{F}$ and every distribution $D$ supported on positive examples, $\mathcal{A}$ runs in polynomial time on NSP–labeled inputs and outputs $\hat{f}$ such that, with probability at least $1 - \delta$, $\mathcal{L}_{\mathrm{NSP}}(\hat{f}; f, D) \leq \epsilon$.

Note that the continuation labels can be highly informative for certain classes. Consider *Conjunctions*: Boolean monomials $f : \{0, 1\}^N \to \{0, 1\}$, e.g., $f(z) = z_2 \wedge \bar{z}_4$. In the conventional classification model, learning Conjunctions is known to require $\Theta(N)$ labeled examples. In the NSP model, one positive example $x \in f^{-1}(1)$ suffices. For each prefix $x_{:k}$, the label $\varphi(x_{:k}, 0)$ equals 0 if and only if the literal $z_{k+1}$ appears in the target; likewise, $\varphi(x_{:k}, 1) = 0$ if and only if the literal $\bar{z}_{k+1}$ appears. Thus, by reading the continuation labels across the $N$ prefixes, the learner recovers exactly which literals are present and hence identifies the target monomial from a single positive NSP example. While the NSP labels may provide a statistical advantage for some classes, we show that in the general case, they are not enough to remove the computational barriers for learning DFAs.

**Main hardness result.** Let $\mathrm{ADFA}^N_{p(N)}$ denote the class of *Boolean Acyclic DFAs* (ADFAs) which contains DFAs with at most $p(N)$ states, whose language is contained in $\{0, 1\}^N$. Kearns & Valiant (1994) show that, for a suitable polynomial $p$, weak PAC learning of $\mathrm{ADFA}_{p(\cdot)}$ in the conventional binary-classification model is as hard as inverting basic cryptographic primitives. We prove that an efficient NSP learner for $\mathrm{ADFA}^N_{p(N)}$ would immediately yield an efficient learner in the standard model and thus the problem remains hard under the same cryptographic assumptions.

**Theorem 4.1.** *Fix $N$ and a polynomial $p(\cdot)$. If $\mathrm{ADFA}^N_{p(N)}$ is efficiently PAC-learnable in the* NSP *setting from positive examples, then $\mathrm{ADFA}^N_{p(N)}$ is efficiently PAC-learnable in the conventional classification (binary-label) setting.*

The proof is in Appendix D. The main technical idea is a construction for ADFAs along with a reduction to the result of Kearns & Valiant (1994). We show that for any Boolean Acyclic DFA in $\mathrm{ADFA}^N_{p(N)}$, there exists another ADFA with at most $N + 1$ additional states such that all but one continuation bit becomes uninformative. Accurately predicting the only informative continuation bit is as hard as learning ADFAs in the conventional classification setting.

**Discussion.** The theorem shows that richer supervision via NSP does not circumvent the computational barrier for learning regular languages: under standard assumptions, there is no polynomial-time weak learner for $\mathrm{ADFA}_{p(\cdot)}$ and hence for DFAs even in the NSP setting. This remains true when the learner receives both positive and negative examples with NSP labels, showing that such additional supervision does not mitigate these barriers. The hardness is *improper*: it rules out efficient learning even when the hypothesis need not be a DFA, and thus applies to neural models trained to match NSP labels.

**Learning with Membership Queries.** In addition to passive examples, we also study *active* access in the NSP model. A conventional membership-query oracle $\mathrm{MQ} : \Sigma^* \to \{0, 1\}$ takes an input string and returns whether it belongs to the target language. In the NSP setting, the oracle $\mathrm{MQ}_{\mathrm{nsp}}(x)$ returns the full vector of $(|\Sigma| + 1)(|x| + 1)$ labels for any $x \in \Sigma^*$. We show that certain simple classes known to be not identifiable in polynomial time with conventional queries become efficiently identifiable with $\mathrm{MQ}_{\mathrm{nsp}}$ with the help of additional labels. However, more generally, we show that some classes of DFAs and Boolean functions remain non-identifiable in polynomial time even with NSP labels from $\mathrm{MQ}_{\mathrm{nsp}}$. See App. E for details.

## 5 Learning with a Language Model Teacher

Given the hardness of learning with random examples or membership queries alone, we now study the learnability of DFAs in a relatively more powerful model based on the information one can conveniently obtain via blackbox access to language models.

**Problem and Assumptions.** Let LM be a language model which induces a distribution over strings $\mathcal{D}_{\mathrm{LM}}$ (correspondingly $\mathcal{D}^{\mathcal{T}}_{\mathrm{LM}}$ for a sampling strategy $\mathcal{T}$). Assume that the support of the distribution $\mathcal{D}_{\mathrm{LM}}$ is a regular language and let $A^\star$ be the DFA that recognizes the support $L_{A^\star}$. Given blackbox

access to such a language model, we would like to find $\hat{A}$ such that $\mathbb{E}_{x \sim \mathcal{D}_{\mathrm{LM}}}[\|f_{\hat{A}}(x) - f_{A^\star}(x)\|_\infty] \leq \epsilon$ with high probability. In this setting, a learner has access to two types of queries: *(i) Membership queries* $\mathrm{MQ}(x) \in \{0, 1\}$ which returns $A^\star(x)$, and *(ii) Generative queries* $\mathrm{Gen}_{\mathcal{D}_{\mathrm{LM}}}(\cdot)$ which takes an input string or prompt $x$ and generates a string $s$ along with NSP labels based on the distribution $\mathcal{D}_{\mathrm{LM}}$ conditioned on the prompt $x$. Note that both these queries can be simulated with blackbox access to the language model (cf. App. G.3).

**Approach.** Broadly, we first sample a set $X$ of size $m$ NSP labeled examples from the distribution $\mathcal{D}_{\mathrm{LM}}$ using Generative queries $\mathrm{Gen}_{\mathcal{D}_{\mathrm{LM}}}(\lambda)$. We will then use an extension of the L$^\star$ algorithm to make use of the membership queries and generative queries to obtain a hypothesis DFA $\hat{A}$ in polynomial time such that $|\hat{A}| \leq |A^\star|$ and $\hat{A}$ is *consistent* with the NSP labels of all $m$ examples. A standard Occam-style generalization bound then yields a PAC-guarantee for the learning problem.

**Preliminaries from L$^\star$.** We briefly recall the notions from L$^\star$ (Angluin, 1987) based on modern treatments (Kearns & Vazirani, 1994) that we use in L$^\star_{\mathrm{nsp}}$ (cf. App. G.1 for a more detailed description). The algorithm maintains finite sets $Q \subseteq \Sigma^*$ of *access words* and $T \subseteq \Sigma^*$ of *test words*, both containing the empty string $\lambda$. Intuitively, $Q$ represents states and $T$ represents distinguishing strings. With respect to a target language $L_{A^\star}$ and any ordering $t_1, \ldots, t_{|T|} \in T$, we can define a row vector of $T$-labels for any string $x$: $\mathrm{row}(x) = [L_{A^\star}(x \cdot t_1), \ldots, L_{A^\star}(x \cdot t_{|T|})] \in \{0, 1\}^{|T|}$. The pair $(Q, T)$ is defined to be separable with respect to language $L_{A^\star}$ if every row vector is unique: $\mathrm{row}(q) \neq \mathrm{row}(q')$ for $q, q' \in Q$. The pair $(Q, T)$ is closed if for every $q \in Q$ and $\sigma \in \Sigma$, there exists $q' \in Q$ such that $\mathrm{row}(q \cdot \sigma) = \mathrm{row}(q')$. When $(Q, T)$ is closed and separable, one can construct a DFA hypothesis with state set $Q$ and transitions based on $T$-labels (see App. G.1 for the construction). A crucial fact about this L$^\star$ framework is that (Lemma G.1) when $(Q, T)$ is closed and separable with respect to language $L_{A^\star}$, then $|Q| \leq |A^\star|$ for the minimal DFA $A^\star$.

Given a set of labeled examples, L$^\star$ starts from $Q = T = \{\lambda\}$ and iteratively adds states and distinguishing strings based on label disagreements, maintaining a key invariant that $(Q, T)$ remains separable. The procedure has two steps: (i) *Closure* (Lemma G.2): if $(Q, T)$ is not closed, use membership queries to update $(Q, T)$ to achieve closure in polynomial time. (ii) *Counterexample processing* (Lemma G.3): once closed, construct a hypothesis DFA $\hat{A}$ and find a *counterexample* $x$ in the training set with $\hat{A}(x) \neq A^\star(x)$. Then, using at most $|x|$ membership queries, one can identify $q' \notin Q$ and $t' \notin T$ so that $(Q \cup \{q'\}, T \cup \{t'\})$ is separable. This step is the backbone of L$^\star$: membership disagreements can be used to obtain new access and test words to refine $(Q, T)$.

In the NSP setting, counterexamples may arise from continuation-labels rather than membership disagreements, so Lemma G.3 does not apply as is. This is where L$^\star_{\mathrm{nsp}}$ departs from L$^\star$: we use a different method to process counterexamples $x \in L_{A^\star}$ using continuation-label disagreements. The next lemma formalizes this update rule for L$^\star_{\mathrm{nsp}}$.

**Lemma 5.1.** *Let $(Q, T)$ be closed and separable, and let $\hat{A}$ be the minimal DFA induced by $(Q, T)$. Suppose there exists a string $x$ with $f_{\hat{A}}(x) \neq f_{A^\star}(x)$. Then one can find $q' \notin Q$ and $t' \notin T$ such that $(Q \cup \{q'\}, T \cup \{t'\})$ is separable, using membership queries and at most one generative query in polynomial time. If a generative query is used, let $y$ denote its output; otherwise set $y = \lambda$. The total number of membership queries is at most $|x| + |y|$, and the running time is polynomial in $|x| + |y| + |Q|$.*

**Proof.** The proof for this Lemma is relatively straightforward. The idea is to convert *any* NSP-label mismatch into an ordinary membership mismatch, and then use the same approach as the classical L$^\star$ update (Lemma G.3). Take $x$ with $f_{\hat{A}}(x) \neq f_{A^\star}(x)$ and let $x_{:n}$ be the first prefix where labels disagree. If the mismatch is the *membership* label, we simply use $x' = x_{:n}$, on which $\hat{A}$ and $A^\star$ disagree so the first case is trivial. If the mismatch is a *continuation* label at some symbol $\sigma$, then there are two complementary situations. *Case (i)* The target says $\sigma$ can never lead to acceptance from $x_{:n}$, but the hypothesis says it can $\varphi_{\hat{A}}(x_{:n}, \sigma) = 1$. In this case, we can find a suffix $s$ from $x_{:n}\sigma$ to an accepting state in the hypothesis automaton and the strings $x' = x_{:n} \cdot s$ is an ordinary membership mismatch. *Case (ii)* The target says $\sigma$ is permissible, but is forbidden according to the hypothesis $\varphi_{\hat{A}}(x_{:n}, \sigma) = 0$. In this case, we can use a single generative query conditioned on $x_{:n}\sigma$ returns a valid continuation $y$; then $x' = x_{:n}\sigma y$ is accepted by the target and rejected by $\hat{A}$. Thus, in both cases, we have a string $x'$ such that $\hat{A}(x') \neq A^\star(x')$ and we can use Lemma G.3 to find desired $q' \notin Q$ and $t' \notin T$. A more detailed version of the proof is in App. G.2.

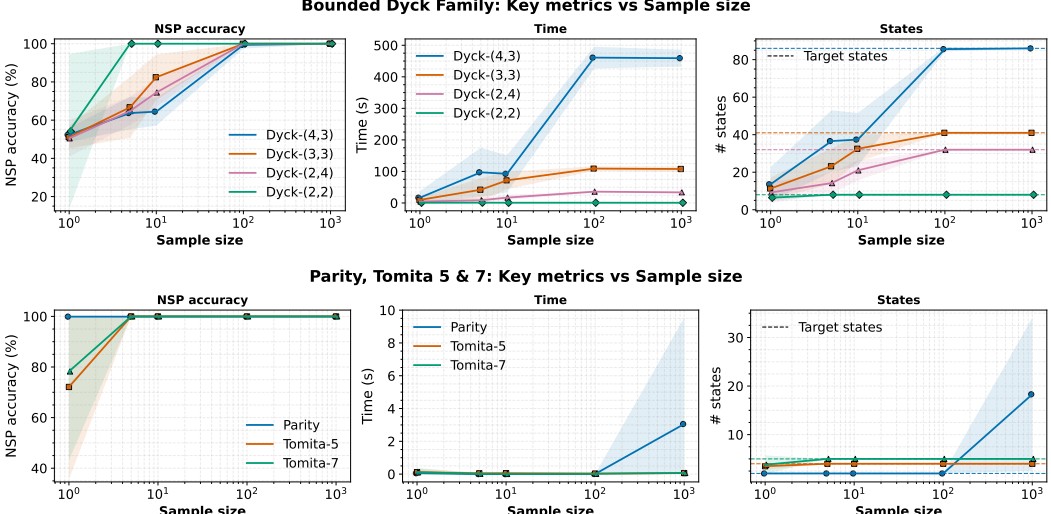

Figure 2: Key metrics for $\mathrm{L}_{\mathrm{nsp}}^{\star}$ across training-set sizes using a Transformer–LM teacher. Each point shows the mean over 10 trials; shaded regions denote standard deviation (see Sec. 6).

**Algorithm.** Given a set of $m$ NSP-labeled example, the $\mathrm{L}_{\mathrm{nsp}}^{\star}$ algorithm works as follows. It starts with $Q = T = \{\lambda\}$ and iteratively updates the pair $(Q, T)$ such that they are always separable. For a hypothesis $\hat{A}$ associated with $(Q, T)$, it finds the first disagreement in the NSP labels with the examples in the training set $X$. Using membership queries and generative queries based on Lemma 5.1, it updates the pair $(Q, T)$ and adds at least one state whenever there is a disagreement with the training examples. The closure step remains the same as in the original $\mathrm{L}^{\star}$ algorithm. The algorithm terminates when $\hat{A}$ induced by $(Q, T)$ is consistent with all the training examples. Since $|Q|$ is guaranteed to be at most $|A^{\star}|$, the algorithm must terminate after at most $|A^{\star}|$ disagreements with the training examples. A pseudocode of the $\mathrm{L}_{\mathrm{nsp}}^{\star}$ is given in Algorithm 1.

**Theorem 5.2.** *Let $A^{\star} \in \mathrm{DFA}_n$ be any minimal DFA with at most $n$ states, and let $\mathcal{D}_{\mathrm{LM}}$ be a distribution over strings whose support is $L_{A^{\star}}$. There exists an algorithm with access to the membership query oracle $\mathrm{MQ}$ and generative query oracle $\mathrm{Gen}_{\mathcal{D}_{\mathrm{LM}}}$ producing NSP labeled examples in $L_{A^{\star}}$, that runs in time polynomial in $n, 1/\epsilon, 1/\delta$ and the length of the largest string produced by the generative query oracle, and with probability at least $1 - \delta$, outputs a DFA $\hat{A}$ such that,*

$$\mathbb{E}_{x \sim \mathcal{D}_{\mathrm{LM}}} [\|f_{\hat{A}}(x) - f_{A^{\star}}(x)\|_{\infty}] \leq \epsilon.$$

**Discussion.** A key benefit of this model of learning is that the guarantee we get is with respect to a desired distribution. For example, if one uses the $\mathrm{L}^{\star}$ algorithm to learn DFAs using positive and negative examples, the target distribution (on negative examples) is often unclear and can be artificial. In practice, for generative models, we often do not have access to the target distribution on *negative* examples. For the purpose of identifying the support of language models, arguably the most relevant distribution is the one induced by the model itself, e.g. if one intends to predict whether the language model will generate erroneous strings.

## 6 EMPIRICAL ANALYSIS

We evaluate the $\mathrm{L}_{\mathrm{nsp}}^{\star}$ algorithm as a tool for extracting DFAs from Transformer language models trained on regular languages. Given NSP-labeled strings from the model, we study how the NSP error $\mathcal{L}_{\mathrm{NSP}}$, the number of identified states, and the running time vary with the size of the training set. When the extracted automaton is not equivalent to the target DFA, we also find strings in the symmetric difference to identify *erroneous examples*. Further, we conduct ablations to analyze the effectiveness and usage of the continuation labels in the NSP setting.

**Tasks.** We consider 11 regular languages spanning 2–86 states: the 6 Tomita grammars (Tomita, 1982), Parity, and 4 bounded Dyck languages. Tomita grammars comprise small DFAs (2–5 states) and is commonly used as a benchmark for automata extraction (Wang et al., 2018; Weiss et al., 2018; Zhang et al., 2024). Parity is the two-state language over $\Sigma = \{0, 1\}$ which contains all strings with an odd number of 1s. Bounded Dyck languages contain well-balanced parentheses up to a fixed depth and are regular. We denote by DYCK-$(n, k)$ the Dyck language with $n$ bracket types and depth at most $k$. The depth of a Dyck string is the maximum number of unclosed brackets in any prefix of the string. These languages have relatively larger number of states: $\sum_{i=0}^{k} n^i + 1$. We use four Dyck instances: DYCK-$(2, 2)$, DYCK-$(2, 4)$, DYCK-$(3, 3)$, and DYCK-$(4, 3)$. Further details of the tasks are in App. H.

**Setup.** For each target language we convert its canonical DFA to a PDFA to generate training strings. At non-final states, probability mass is split uniformly among transitions that avoid the dead state; at final states, generation terminates with probability $t$ (otherwise, the next symbol is sampled as above). For each language, we choose $t$ so that the empirical expected length is below 40.

**Model training.** We train Transformers as next-token predictors on sequences of the form [BOS] $s_1$ [EOS] $s_2$ [EOS] $\cdots$ with context window 250. Models use 8 layers and width 512, optimized with AdamW for up to 40k steps with early stopping. Our evaluation focuses on the support of the first string produced after [BOS]; the concatenated format matches standard training and provides additional learning signal.

**DFA extraction.** We use the trained Transformers to generate training sets of various sizes to evaluate the $\mathrm{L}^\star_{\mathrm{nsp}}$ algorithm. We create training sets with strings of length up to 80 that are generated and labeled using min-$p$ sampling with threshold $p = 0.05$ as described in Sec. 2.2. The Transformer model serves as both membership and generative query oracle for $\mathrm{L}^\star_{\mathrm{nsp}}$. We evaluate sample sizes $\{1, 5, 10, 100, 1000\}$; for each size we run 10 independent trials and report means and standard deviations. For very small targets (e.g., Tomita), even a single positive example can be informative.

**Results.** Figure 2 summarizes NSP accuracy, running time, and the number of extracted states for representative tasks. On Tomita, when teacher models are well trained, $\mathrm{L}^\star_{\mathrm{nsp}}$ recovers the target DFA quickly (often within a second; bottom row). For models that are not perfectly trained, such as for Parity, the algorithm extracts a much larger DFA and takes longer, depending on the number of target states. Note that since $\mathrm{L}^\star_{\mathrm{nsp}}$ adds a state only after finding a distinguishing string, the number of extracted states is always at most the number of states in the target DFA that recognizes the language model's support. Thus, even when we recover more states than the target DFA, the result is faithful to the language teacher's support.

Bounded Dyck languages have relatively much larger number of states (8–86 states). As shown in the top row of Figure 2, Dyck tasks naturally require more samples than Tomita, yet $\mathrm{L}^\star_{\mathrm{nsp}}$ converges to the target DFA and achieves near-perfect NSP accuracy within 100 examples. Our ablation experiments in App. H.4 indicate that the continuation labels are heavily used and play a crucial role in identifying the states of the target (see Table 3).

**Identifying erroneous examples.** When the learned DFA $\hat{A}$ is not equivalent to the target DFA $A^\star$, we construct the product DFA $B$ which recognizes the strings in the symmetric difference of the two languages $L(B) = L(\hat{A}) \triangle L(A^\star)$. We use a BFS-like approach to identify several erroneous examples for the language model. Table 2 illustrates some erroneous examples for Bounded Dycks, Parity, and Tomita-5 language. Fig. 6 and 7 depict the extracted automaton for Parity and Tomita-5; the ones for DYCK-$(2, 2)$ and DYCK-$(3, 3)$ are too large to be visually informative. Note that these models were not intentionally trained to fail, and all the examples generated by the language models were in their respective target languages. The DFAs extracted by $\mathrm{L}^\star_{\mathrm{nsp}}$ were based on a few disagreements in the NSP labels of the generated strings. Training the language models for longer avoids such errors for synthetic languages of this scale. Note that the Transformer models used for Tomita-5 and Dyck languages in Figure 2 (well-trained) and Table 2 (imperfect) are different. See App. H.2 for further details.

**NSP Label vs Binary Label Ablations.** To assess the value of NSP labels, we compare binary labels (classical $\mathrm{L}^\star$) with NSP labels ($\mathrm{L}^\star_{\mathrm{nsp}}$ extension) on six languages. We sample strings from the model's *untruncated* distribution (actual next-token probabilities) and label each as positive if it lies in the min-$p$ truncated support $\mathcal{D}_{\mathrm{LM}}^{\mathcal{T}}$ used elsewhere, negative otherwise. Naturally, most ($\approx 99.5\%$)

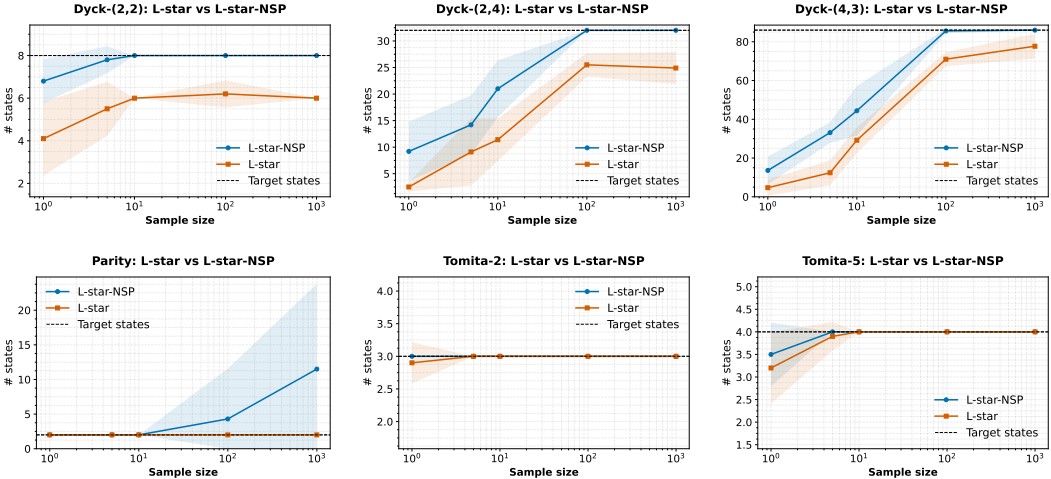

Figure 3: Comparison between binary labels ($L^\star$) and NSP labels ($L^\star_{nsp}$) by including negative examples by sampling from the untruncated distribution of the LM. Each point in the plots is the mean value over 10 trials; shaded regions show standard deviation. See Sec. 6 for details.

of the samples are positive; nevertheless, even positive strings serve as counterexamples for $L^\star$ and using any natural variant of the $\mathcal{D}_{LM}$ variant will have few negative examples.

**Results.** Figure 3 plots the number of extracted states versus sample size (means over 10 runs). On bounded Dyck languages, $L^\star_{nsp}$ reaches the target DFA with $\approx 10$ examples for DYCK-$(2, 2)$ and $\approx 100$ for DYCK-$(2, 4)$ and DYCK-$(4, 3)$, whereas $L^\star$ fails to recover the target even with $10^3$ samples. For small Tomita DFAs, a single counterexample suffices, so both approaches perform similarly. For *Parity*, although the target DFA has 2 states, the teacher's support DFA is larger; with NSP labels, $L^\star_{nsp}$ identifies this larger support (enabling the discovery of erroneous strings), while binary labels yield no disagreements and thus no additional states. Because almost all samples are positive, classification accuracy is uninformative; predicting 1 leads to near perfect accuracy and thus the extracted state count is the meaningful signal. Note that the claim here isn't that $L^\star_{nsp}$ is superior (it is a direct extension of $L^\star$ itself). The primary goal is to assess whether the additional labels are informative and the results indicate that leveraging NSP labels can be sample efficient for problems where the natural distribution primarily has positive examples.

# 7 FUTURE WORK AND LIMITATIONS

A natural question that remains open is the efficient learnability of DFAs with membership (MQ) and equivalence queries (EQ) in the NSP setting. We show that with membership queries and two types of equivalence queries, DFAs are exactly learnable and discuss barriers behind obtaining a standard MQ+EQ algorithm in App. G.4. Prop. 3.1 shows that NSP labels are sufficient and a teaching (or characteristic) set exists. However, the size of such a set obtained from Prop. 3.1 is likely to be quite loose and could possibly be improved.

**Limitations.** Even though the $L^\star_{nsp}$ algorithm is polynomial-time, since it is built upon the $L^\star$ framework, some of the same limitations apply, which make it difficult to scale the approach to practical language models. In particular, when the target DFA (language model support) has a large number of states, the algorithm is quite slow. We observe that when models are poorly trained, the underlying DFA typically has thousands of states. The $L^\star_{nsp}$ algorithm identifies about 1k states and the closure step is slow due to the $|Q||T||\Sigma|$ time complexity (discussed in more detail in App. H.3). Further, the factor of $|\Sigma|$ may not play a significant role for synthetic languages but has immediate consequences for language models trained on text which have a large vocabulary. An interesting future work would be to develop more efficient system-level improvements to speed up the algorithm to make it applicable to relatively more practical language models.

ACKNOWLEDGEMENTS

We would like to thank Dana Angluin, Alex Clark, Charles London, and anonymous reviewers for their valuable feedback and constructive suggestions on this work.

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

# Contents

## A   USAGE OF LARGE LANGUAGE MODELS

We used publicly available AI tools (LLMs) to improve the flow and clarity of the text in various places. LLMs were used to rephrase certain text to polish it and/or make it more concise. AI tools have also aided in writing the code for the experiments in the paper.

## B   FAQS

**(1)** *If $\mathrm{L}^{\star}_{\mathrm{nsp}}$ is an efficient polynomial time algorithm, why shouldn't we directly apply it to large language models (LLMs) to obtain a DFA that captures their support?*

There are a few nontrivial obstacles in applying the algorithm to LLMs. Firstly, the target language for an LLM could have a huge number of states, and currently, it is difficult to use the algorithm for targets beyond the order of a few thousand states. Secondly, while the complexity of the algorithm is polynomial, it still scales linearly with the size of the vocabulary, and LLMs have vocabularies that are in the order of tens of thousands. Lastly, the support of LLMs may not be regular, which could violate the main assumption, and the guarantees need not apply. We believe this work makes progress towards the larger goal of capturing the support of language models, but it is practically limited, similar to other existing methods for automata extraction.

**(2)** *To identify erroneous examples, can we not sample from a language model several times and then check if any of them are incorrect?*

In theory, one can sample repeatedly to identify erroneous strings in the support. We argue that better algorithms for this problem can make the identification of erroneous strings much more efficient. For instance, for the parity language, sampling 1k strings from a seemingly well-trained model does not seem to produce any erroneous examples. However, the DFA extracted by leveraging NSP labels provides us with numerous erroneous examples. We found that $L^\star_{\text{nsp}}$ could extract a DFA within 10 seconds using 1k generations from the language model, which could then be used to produce about 1k erroneous strings within 16 seconds (see App H.2 for details). While the current approach isn't scalable beyond such synthetic problems, further improvements could help make progress in more efficient identification of DFAs and, consequently, identification of erroneous examples.

**(3)** *On what kind of problems is it more beneficial to use $L^\star_{\text{nsp}}$ as opposed to directly using $L^\star$?*

$L^\star_{\text{nsp}}$ is a direct extension of $L^\star$ that exploits NSP labels when available. It can be more sample-efficient in positive-only or highly imbalanced settings typical of generative models. Empirically, on bounded Dyck languages, $L^\star_{\text{nsp}}$ recovered the target DFA with $\approx$10–100 examples, whereas without NSP labels, $L^\star$ did not with $10^3$ samples (Fig. 3); ablations also show the algorithm heavily uses continuation labels on Dyck tasks (see Table 3 and Sec. H.4).

**(4)** *Does the hardness result imply that learning with random examples alone will likely fail?*

One should take note that the hardness results are generally for the worst-case setting. It is helpful to gain a formal understanding of the strengths and weaknesses of a problem but they do not immediately imply that learning will fail most of the times. The result indicates that efficient distribution-independent PAC-learning algorithms are not feasible under standard cryptographic assumptions. However, in practice, using scalable heuristic-based methods is often effective. Our goal was to characterize the learnability in the NSP setting to understand the power of the additional supervision available in the NSP setting and it implies that they are not powerful enough to mitigate certain computational barriers.

## C  PRELIMINARIES

We define Probabilistic DFAs (PDFAs) formally here. Our result on learning with membership and generative queries (Theorem 5.2) has direct implications on learning the support of PDFAs. We also use PDFAs to generate strings for training Transformer language models.

**Probabilistic DFAs (PDFAs).** A probabilistic DFA is a DFA equipped with a stochastic emission rule. Formally, a PDFA is a tuple

$$\mathcal{P} = (Q, \Sigma, \delta, q_0, F, \pi),$$

where $(Q, \Sigma, \delta, q_0, F)$ is a DFA and, for each $q \in Q$, $\pi(\cdot \mid q)$ is a probability distribution on $\Sigma \cup \{\texttt{[EOS]}\}$. At state $q$, the generator samples $w \in \Sigma \cup \{\texttt{[EOS]}\}$ according to $\pi(\cdot \mid q)$; if $w \in \Sigma$ the next state is $\delta(q, w)$, and if $w = \texttt{[EOS]}$ the sequence terminates. We say a string $x = w_1 \cdots w_N \in \Sigma^*$ is in the *support* of $\mathcal{P}$ if

$$\pi(w_1 \mid q_0) \cdot \pi(w_2 \mid \delta(q_0, w_1)) \cdots \pi(w_N \mid \delta(q_0, x_{:N-1}))\pi(\texttt{[EOS]} \mid \delta(q_0, x)) > 0.$$

The DFA that accepts exactly this support is the *support DFA* of $\mathcal{P}$.

**NSP labels from PDFAs.** In a PDFA $\mathcal{P}$, the NSP labels coincide with positivity of the local emission probabilities:

$$\varphi(y, \sigma) = \mathbb{I}\{\pi(\sigma \mid \delta(q_0, y)) > 0\}, \qquad L(y) = \mathbb{I}\{\pi(\texttt{[EOS]} \mid \delta(q_0, y)) > 0\}.$$

Thus, the NSP labelling oracle exposes exactly the admissible next symbols and termination at each prefix; for a PDFA this recovers the (untruncated) support of $\mathcal{P}$.

The following is an elementary but fundamental fact about minimal DFAs due to the Myhill-Nerode Theorem that is at the heart of many of our proofs and constructions.

**Lemma C.1** (DFA basic fact). *Let $A$ be a minimal DFA recognizing $L_A$. For any two distinct strings $x, y \in \Sigma^*$, if there exists a suffix $s \in \Sigma^*$ such that $L_A(x \cdot s) \neq L_A(y \cdot s)$ then the strings $x$ and $y$ lead to two distinct states in the DFA $A$, i.e., $\delta_A(q_0, x) \neq \delta_A(q_0, y)$. If no such suffix exists, then they lead to the same state $\delta_A(q_0, x) = \delta_A(q_0, y)$.*

In a minimal DFA, distinct states are pairwise distinguishable by some continuation; conversely, strings with indistinguishable residual languages must reach the same state. This is the Myhill–Nerode characterization of minimality.

**Lemma C.2.** *Let the dead state $q_{\text{dead}}$ be a state such that $q_{\text{dead}} \notin F$ and $\delta(q_{\text{dead}}, \sigma) = q_{\text{dead}}$ for all $\sigma \in \Sigma$. Then, a minimal DFA has at most one dead state.*

Assume there are two dead states. Since every suffix from a dead state is rejected, there cannot be a suffix that is accepted by one state and rejected by another. Further, by definition, both of them are non-final. Hence, they are Myhill–Nerode equivalent and must be merged in a minimal DFA.

**Lemma C.3.** *Let $\text{DFA}_n$ be the class of DFAs over a fixed alphabet $\Sigma$ with at most $n$ states. Then $\log |\text{DFA}_n| = \mathcal{O}(n \log n)$.*

This is a well-known result (De la Higuera, 2010) and can be proved in the following way. The proof follows from the fact that for a DFA with at most $n$ states, there are $n^{|\Sigma| \cdot n}$ and each state can either be an accept or reject state adds a factor of $2^n$, and one of the states can be an initial state. Hence, the total number of possible DFAs is $2^n \cdot n^{|\Sigma| \cdot n + 1}$. Thus, $\log |\text{DFA}_n| = \mathcal{O}(n \log n)$.

# D  HARDNESS OF NSP LEARNING

We relate learnability in the NSP setting to standard PAC learning for acyclic DFAs that accept only strings of a fixed length. Throughout this section, we work over the binary alphabet $\Sigma = \{0, 1\}$.

**Notation for classes.** For $N \in \mathbb{N}$ and a polynomial $p(\cdot)$, define

$$\text{ADFA}_{p(N)}^N := \{A : A \text{ is a DFA with at most } p(N) \text{ states and } L_A \subseteq \{0, 1\}^N\},$$

and write $\text{ADFA}_{p(\cdot)} := \bigcup_{N \geq 1} \text{ADFA}_{p(N)}^N$. As in the context, $\text{DFA}_n$ denotes all DFAs with at most $n$ states.

**Background.** Kearns & Valiant (1994) show that, for a suitable polynomial $p$, weak PAC learning of $\text{ADFA}_{p(\cdot)}$ in the conventional classification (binary-label) model is as hard as inverting certain cryptographic functions.

**Theorem D.1** (Kearns & Valiant (1994)). *There exists a polynomial $p(\cdot)$ such that the problems of inverting RSA, factoring Blum integers, etc., are probabilistic polynomial-time reducible to weakly learning $\text{ADFA}_{p(\cdot)}$ in the standard PAC setting.*

We prove that an efficient PAC algorithm for $\text{ADFA}_{p(N)}^N$ in the NSP setting would yield an efficient PAC algorithm for $\text{ADFA}_{p(N)}^N$ in the conventional classification setting. Together with Theorem D.1, this implies cryptographic hardness for NSP learning of these Boolean Acyclic DFAs and consequently the general class of DFAs.

## D.1  BOOLEAN ACYCLIC DFAS

Fix $N \geq 1$ and a target DFA $A = (Q, \Sigma, \delta, q_0, F)$ with $L_A \subseteq \{0, 1\}^N$. We first record a basic structural property of *minimal* DFAs for fixed-length languages.

**Lemma D.2** (Unique depth in minimal $\text{ADFA}_{p(N)}^N$). *If $A$ is minimal and $L_A \subseteq \{0, 1\}^N$, then every state $q \in Q \setminus \{q_{\text{dead}}\}$ is reachable by strings of exactly one length $\ell(q) \in \{0, 1, \ldots, N\}$. Consequently, every transition increases depth by one: if $\delta(q, \sigma) = q'$ and $q \neq q_{\text{dead}}$, then $\ell(q') = \ell(q) + 1$. Acceptance occurs only at depth $N$.*

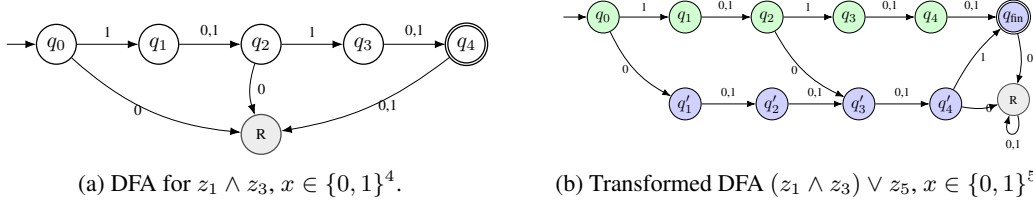

(a) DFA for $z_1 \wedge z_3$, $x \in \{0, 1\}^4$.

(b) Transformed DFA $(z_1 \wedge z_3) \vee z_5$, $x \in \{0, 1\}^5$.

Figure 4: Transformation from Section D.1. In (b), green states are from the original DFA; blue states $q_1', \ldots, q_4'$ ensure every prefix of length $< N$ has NSP labels $[1, 1, 0]$, and $q_4'$ routes to accept on input 1 (to the dead state on 0).

*Proof.* Let $q \in Q \setminus \{q_{\text{dead}}\}$ be reachable both by a string of length $\ell$ and by a string of length $\ell'$ with $\ell < \ell'$. The residual language at $q$ is $R(q) := \{ s \in \Sigma^* : \delta(q, s) \in F \}$. If $q$ is reached after $\ell$ symbols, then every $s \in R(q)$ must have length exactly $N - \ell$; if $q$ is reached after $\ell'$ symbols, then every $s \in R(q)$ must have length exactly $N - \ell'$. Since $N - \ell \neq N - \ell'$, these sets are disjoint; hence $R(q)$ must be empty, contradicting $q \neq q_{\text{dead}}$ in a minimal DFA. Thus, each non-dead state has a unique depth $\ell(q)$. Any transition consumes one symbol, so $\ell(q') \leq \ell(q) + 1$; equality must hold by uniqueness of depth. Finally, if $q \in F$ had $\ell(q) \neq N$, then $L_A$ would contain strings of length other than $N$, contrary to the assumption. $\square$

By Lemma D.2, we may index non-dead states by their unique depth $\ell(q) \in \{0, \ldots, N\}$ (the dead state has no depth).

**Padding by one bit.** We construct from $A$ a DFA $A^\oplus$ over length-$N{+}1$ inputs whose NSP labels are uninformative before depth $N$, while at depth $N$ the continuation bit for symbol 0 recovers $A$'s label.

**Lemma D.3** (Padded Construction). *From a Boolean Acyclic DFA $A$ we can construct, in time polynomial in $|A| + N$, a DFA $A^\oplus$ with $L_{A^\oplus} \subseteq \{0, 1\}^{N+1}$ and*

$$u \cdot b \in L_{A^\oplus} \iff (A(u) = 1) \text{ or } (b = 1), \qquad u \in \{0, 1\}^N, \ b \in \{0, 1\}. \tag{2}$$

*Moreover, for every prefix $y$ with $|y| < N$,*

$$\big(\varphi(y, 0), \ \varphi(y, 1), \ L_{A^\oplus}(y)\big) = (1, 1, 0),$$

*and for every $u \in \{0, 1\}^N$,*

$$\big(\varphi(u, 0), \ \varphi(u, 1), \ L_{A^\oplus}(u)\big) = \big(A(u), 1, 0\big).$$

*The construction adds at most $N{+}1$ states.*

*Proof.* Introduce new states $q_1', \ldots, q_N'$ and a new accepting state $q_{\text{fin}}$. For $1 \leq i < N$ set

$$\delta_{A^\oplus}(q_i', 0) = q_{i+1}', \qquad \delta_{A^\oplus}(q_i', 1) = q_{i+1}',$$

and at $i = N$ set

$$\delta_{A^\oplus}(q_N', 1) = q_{\text{fin}}, \qquad \delta_{A^\oplus}(q_N', 0) = q_{\text{dead}}.$$

From $q_{\text{fin}}$ send both symbols to $q_{\text{dead}}$ (so acceptance occurs only at length $N{+}1$).

Now modify $A$ to create $A^\oplus$ as follows. For each non-dead state $q$ with $\ell(q) < N$ and each $\sigma \in \{0, 1\}$:

- If $\delta_A(q, \sigma) = q_{\text{dead}}$ in $A$, set $\delta_{A^\oplus}(q, \sigma) = q_{\ell(q)+1}'$ (redirect the dead transition into the chain).

- Otherwise set $\delta_{A^\oplus}(q, \sigma) = \delta_A(q, \sigma)$.

For each state $q$ at depth $N$ set

$$\delta_{A^\oplus}(q, 1) = q_{\text{fin}}, \qquad \delta_{A^\oplus}(q, 0) = \begin{cases} q_{\text{fin}}, & q \in F_A, \\ q_{\text{dead}}, & q \notin F_A. \end{cases}$$

See Figure 4 for a simple example construction for a Boolean Acyclic DFA computing a conjunction.

All transitions out of $q_{\text{dead}}$ point to $q_{\text{dead}}$, i.e., $\delta_{A\oplus}(q_{\text{dead}}, \sigma) = q_{\text{dead}}$ for both symbols. (If $A$ recognizes the empty language, then $Q = \{q_{\text{dead}}\}$; in this case we additionally introduce a fresh start state $\tilde{q}_0$ of depth 0 with $\delta_{A\oplus}(\tilde{q}_0, 0) = \delta_{A\oplus}(\tilde{q}_0, 1) = q'_1$ and take $\tilde{q}_0$ as the start state; the conclusions below still hold.)

By construction, acceptance can occur only at $q_{\text{fin}}$ after exactly $N+1$ symbols, so $L_{A\oplus} \subseteq \{0,1\}^{N+1}$. The rule at depth $N$ gives (2). For any prefix $y$ with $|y| < N$, either an original transition still has an accepting continuation, or a dead transition is redirected to the chain $q'_{|y|+1} \to \cdots \to q'_N \to q_{\text{fin}}$ (taking the last symbol 1). Hence $\varphi(y, 0) = \varphi(y, 1) = 1$ and $L_{A\oplus}(y) = 0$ for any $y$ with $|y| < N$. At depth $N$, for every $u \in \{0,1\}^N$ our construction enforces

$$\delta_{A\oplus}\big(\delta_{A\oplus}(q_0, u), 1\big) = q_{\text{fin}} \quad \text{and} \quad \delta_{A\oplus}\big(\delta_{A\oplus}(q_0, u), 0\big) = \begin{cases} q_{\text{fin}}, & \delta_A(q_0, u) \in F_A, \\ q_{\text{dead}}, & \delta_A(q_0, u) \notin F_A, \end{cases}$$

so $\varphi(u, 1) = 1$, $\varphi(u, 0) = A(u)$, and $L_{A\oplus}(u) = 0$, which is exactly

$$\big(\varphi(u, 0),\ \varphi(u, 1),\ L_{A\oplus}(u)\big) = \big(A(u),\ 1,\ 0\big).$$

$\square$

**Reduction from standard learning to NSP learning.** Let $D$ be any distribution on $\{0,1\}^N$. Define the *padded* distribution $D^\oplus$ on $\{0,1\}^{N+1}$ by sampling $u \sim D$ and returning $x := u \cdot 1$. By (2), $x \in L_{A\oplus}$ for every $u$, so $D^\oplus$ is supported on positive examples as required by the NSP setting. Moreover, given a labeled standard example $(u, y)$ with $y = A(u)$, the full NSP label vector $f_{A\oplus}(x)$ for $x = u \cdot 1$ is computable from $(u, y)$:

$$\begin{aligned} \text{for } 0 \leq \ell < N: \quad & \big(\varphi(x_{:\ell}, 0), \varphi(x_{:\ell}, 1), L(x_{:\ell})\big) = (1, 1, 0), \\ \text{for } \ell = N: \quad & \big(\varphi(x_{:N}, 0), \varphi(x_{:N}, 1), L(x_{:N})\big) = (y, 1, 0), \\ \text{for } \ell = N+1: \quad & \big(\varphi(x_{:N+1}, 0), \varphi(x_{:N+1}, 1), L(x_{:N+1})\big) = (0, 0, 1). \end{aligned}$$

Here $x_{:\ell}$ denotes the length-$\ell$ prefix of the padded string $x$.

**Theorem 4.1.** *Fix $N$ and a polynomial $p(\cdot)$. If $\mathrm{ADFA}_{p(N)}^N$ is efficiently PAC-learnable in the NSP setting from positive examples, then $\mathrm{ADFA}_{p(N)}^N$ is efficiently PAC-learnable in the conventional classification (binary-label) setting.*

*Proof.* Let $\mathcal{A}_{\text{NSP}}$ be an efficient NSP learner for $\mathrm{ADFA}_{p(N)}^N$. From i.i.d. labeled samples $(u^{(i)}, y^{(i)})$ with $y^{(i)} = A(u^{(i)})$, form positive NSP examples $\big(x^{(i)}, f_{A\oplus}(x^{(i)})\big)$ where $x^{(i)} = u^{(i)} \cdot 1$ using the rule above, and feed them to $\mathcal{A}_{\text{NSP}}$. Let $\hat{f}$ be the returned predictor so that, with probability at least $1 - \delta$,

$$\mathcal{L}_{\text{NSP}}\big(\hat{f};\, f_{A\oplus},\, D^\oplus\big) = \mathbb{E}_{u \sim D}\Big[\big\|\hat{f}(u \cdot 1) - f_{A\oplus}(u \cdot 1)\big\|_\infty\Big] \leq \epsilon.$$

Define a standard classifier $h : \{0,1\}^N \to \{0,1\}$ by

$$h(u) := \text{ the bit predicted by } \hat{f} \text{ for } \varphi(x_{:N}, 0) \text{ on } x = u \cdot 1.$$

Whenever $\big\|\hat{f}(x) - f_{A\oplus}(x)\big\|_\infty = 0$, Lemma D.3 gives $h(u) = A(u)$. Therefore

$$\Pr_{u \sim D}\big[h(u) \neq A(u)\big] \leq \Pr_{u \sim D}\Big[\big\|\hat{f}(u \cdot 1) - f_{A\oplus}(u \cdot 1)\big\|_\infty = 1\Big] \leq \epsilon,$$

yielding an efficient PAC learner in the standard setting. The state complexity of $A^\oplus$ is at most $p(N) + N + 1$, preserving polynomial time complexity. $\square$

**Corollary D.3.1** (Cryptographic hardness for NSP learning of fixed-length acyclic DFAs)**.** *Under the assumptions of Theorem D.1, there is no polynomial-time weak learner for $\mathrm{ADFA}_{p(\cdot)}$ in the NSP setting. Otherwise, Theorem 4.1 would yield a polynomial-time weak learner in the standard setting, contradicting Theorem D.1.*

## D.2 BOOLEAN FORMULAS

Exactly the same padding idea applies to Boolean formulas. Let $F$ be any class of formulas $f : \{0,1\}^N \to \{0,1\}$ and define

$$f'(z_1, \ldots, z_{N+1}) := f(z_1, \ldots, z_L) \vee z_{N+1}.$$

Given a labeled standard example $(u, y)$ with $y = f(u)$, set $x := u \cdot 1$. The NSP labels for the positive string $x$ under $f'$ are then computable from $(u, y)$:

$$\text{for } \ell < N : \ (1,1,0), \qquad \text{for } \ell = N : \ (y,1,0), \qquad \text{for } \ell = N+1 : \ (0,0,1),$$

with the same ordering (continuations first, then membership) as in §2. Thus an efficient NSP learner for $F' := \{f' : f \in F\}$ yields, by reading the depth-$N$ continuation bit for symbol 0, an efficient PAC learner for $F$ in the standard setting. In particular, cryptographic hardness results for learning formulas (e.g., via $\text{NC}^1$) carry over to the NSP setting by this reduction.

# E HARDNESS OF LEARNING WITH MEMBERSHIP QUERIES ONLY

**Definition.** A conventional membership query oracle $\text{MQ} : \Sigma^* \to \{0,1\}$ takes an input string and returns whether it belongs to the target language or not. In the NSP setting, the membership query oracle $\text{MQ}_{\text{nsp}} : \Sigma^* \to \{0,1\}^{(|\Sigma|+1)(|x|+1)}$ returns all the NSP labels for an input string $x$. Since it contains that membership label of the input in its label, it is strictly more powerful than the conventional MQ oracle.

To understand how additional labels could provide more information, consider the following. If for any pair of prefixes or strings $x, y$, the $|\Sigma|$ continuation labels $\varphi(x), \varphi(y)$ differ, then it implies that they lead to two distinct states in the target DFA. This is because disagreement in the continuation label implies that there exists a suffix $s$ such that $L(x \cdot s) \neq L(y \cdot s)$ and by Lemma C.1, they must lead to two different states.

There are some classes of functions that can be efficiently identified by $\text{MQ}_{\text{nsp}}$ but not by conventional MQ. For instance, consider the class of singleton Boolean functions $\mathcal{F}_1$ over $\{0,1\}^N$ which contains $2^N$ functions that accept exactly 'one' Boolean input of length $N$. Such functions cannot be identified by conventional membership queries in polynomial time (see Angluin (1988) for a general characterization). The main idea is that no matter which input in $\{0,1\}^N$ a learner queries, an adversary can decide to always return 0 as the label until the learner queries $2^N - 1$ inputs.

**Learning Singletons with** $\text{MQ}_{\text{nsp}}$. The membership query oracle in the NSP setting is more powerful in the sense that such singleton Boolean functions can be identified easily in polynomial time. To see how, consider the following procedure: Let $x^* \in \{0,1\}^N$ be the only string accepted by the target $f^* : \{0,1\}^N \to \{0,1\}$. A learner first queries $\text{MQ}_{\text{nsp}}(x)$ any $x \in \{0,1\}^N$ and checks the $|\Sigma|$ continuation labels for the first index corresponding to empty prefix $\lambda$. That indicates the first bit in the string accepted by the target function. Let $x_1^*$ be the first bit of the target string, the learner then queries any string starting with $x_1^*$ and obtains the second bits. Similarly, it can iteratively query $\text{MQ}_{\text{nsp}}$ and obtain the string $x^*$ with at most $N + 1$ $\text{MQ}_{\text{nsp}}$ queries.

While the membership query oracle in the NSP setting $\text{MQ}_{\text{nsp}}$ is strictly more powerful than the one in the conventional classification setting, we show that there are DFAs in the class $\text{DFA}_n$ that cannot be efficiently identified with membership queries only in the NSP setting. Similar to the case of hardness of learning with DFAs, we will construction functions where the NSP labels are uninformative and the problem becomes as hard as learning with the conventional membership query oracle.

**Suffix Language family.** Consider the following family of languages. Let $\Sigma = \{0,1\}$ (the argument applies to any $\Sigma$ with $|\Sigma| \geq 2$). Let $S = \{0,1\}^{N/2}$ be the set of Boolean strings of length exactly $N/2$. The suffix language family $\mathcal{L}_S$ contains $2^{N/2}$ languages, where for each $s \in \{0,1\}^{N/2}$, the language $L_s \in \mathcal{L}_S$ only accepts strings which end with the suffix $s$.

Each of the language $L_s \in \mathcal{L}_S$ can be represented by a DFA of size at most $N/2 + 1$. For any suffix or string $s$, create a state corresponding to each prefix of $s = s_0 \cdot s_1 \cdots s_{N/2}$ including the empty prefix $s_0 = \lambda$ which serves as the start state. Define transitions $\delta(s_{:k}, s_{k+1}) = s_{:k+1}$. For every

other transition, if the symbol is $s_1$, then they go to the state $s_{:1}$ or else they go to $s_0$. From the first state, the shortest string that leads to the accept state is of length $N/2$. For every other state $q_i$ in $q_1, \ldots, q_{N/2}$, the shortest accepting suffix is of length $\frac{N}{2} - i$.

**Proposition E.1.** *The class of Suffix languages $\mathcal{L}_S$ cannot be identified in polynomial time with membership queries* $\mathrm{MQ}_{\mathrm{nsp}}$ *in the* NSP *setting.*

*Proof.* A key characteristic of any of suffix language $L_s$ is that, an accepting string exists for any prefix $x$. For any string $x$, the string $x \cdot s$ is in the language $L_S$. Thus, the continuation label for any prefix of any string will always be $(\varphi(x, 0), \varphi(x, 1)) = (1, 1)$. Hence, the continuation labels are not informative. The only informative signals are the membership labels.

In the NSP setting, the oracle $\mathrm{MQ}_{\mathrm{nsp}}$ will provide the membership labels of every prefix. Suppose a learner makes $m$ $\mathrm{MQ}_{\mathrm{nsp}}$ queries where the maximum length of the queried input strings is $k$. Then each query eliminates at most $(k + 1) - \frac{N}{2} \leq k$ suffixes out of $2^{N/2}$ possible suffixes if all the membership labels are 0. Thus, in the worst case, the learner can eliminate at most $O(mk)$ suffixes or functions from the class. If both the number of queries $m$ and input length $k$ are polynomial in $N$, then they cannot identify the target $s$. Hence, either the number of queries must be exponential or the learner must use exponential computational steps. $\square$

Since the suffix language with suffixes of size $n/2$ can be represented with DFAs with at most $n$ states, Prop. E.1 immediately implies that the class $\mathrm{DFA}_n$ cannot be identified in polynomial time with membership queries alone in the NSP setting even with the richer set of labels.

**Boolean functions and Acyclic DFAs.** A similar argument applies to the class of Boolean functions as well. Consider the class of functions $\mathcal{F}_{2^N - 1}$ over $\{0, 1\}^N$ which accept all bit strings of length $N$ except one. It is straightforward to see that every function in that class can also be represented by Boolean Acyclic DFAs with exactly $N + 2$ states.

For such a class, the continuation labels will be unhelpful for all but one prefix of length $N - 1$. The same line of argument as earlier shows that each membership query can eliminate at most 1 function from the class. An adversary can decide to return membership labels 0 for every input query and choose the function based on the inputs queried by the learner, and hence in the worst case, the learner must make $2^{N-1} - 1$ queries before it can identify the target functions.

This implies that certain classes of Boolean functions as well as the class $\mathrm{ADFA}_n$ cannot be identified with polynomial number of $\mathrm{MQ}_{\mathrm{nsp}}$ queries in the NSP setting.

## F    EQUIVALENCE QUERIES AND IDENTIFIABILITY

Let $\hat{A}$ and $A^\star$ be a hypothesis and target DFA, respectively. In the NSP setting, a valid Equivalence Query oracle $\mathrm{EQ}_{\mathrm{nsp}}^+(\hat{A}; A^\star)$ with respect to target $A^\star$ should take a hypothesis $\hat{A}$ as input and output 'equivalent' if $L_{\hat{A}} = L_{A^\star}$ or else it should return a counterexample $x \in L_A^\star$ such that $f_{\hat{A}}(x) \neq f_{A^\star}(x)$. In other words, the counterexample is a string that is accepted by the target DFA $A^\star$ but disagrees with $\hat{A}$ on at least one of the NSP labels.

Since the Equivalence query oracle can only produce strings accepted by $A^\star$, it is not immediately clear whether a counterexample will always exist when $L_{\hat{A}} \neq L_{A^\star}$. We show that positive examples with NSP labels are sufficient in the sense that for any pair of DFAs $\hat{A}, A^\star$ such that $L_{A^\star} \neq \emptyset$, there is always a counterexample if $L_{\hat{A}} \neq L_{A^\star}$. The following result is crucial for exact learning of DFAs with membership and equivalence queries to be feasible.

**Proposition 3.1.** *Let $A \neq A^\star$ be minimal DFAs with $L_{A^\star} \neq \emptyset$. Then there exists $x \in L_{A^\star}$ such that $f_A(x) \neq f_{A^\star}(x)$. Equivalently, the oracle $\mathrm{EQ}(A; A^\star)$ is well-defined: it either returns "equivalent" or a positive counterexample $(x, f_{A^\star}(x))$.*

*Proof.* Since $A \neq A'$, there exists a string $z \in \Sigma^*$ with $A(z) \neq A'(z)$. If $A'(z) = 1$, we may take $x = z$; the NSP vectors then disagree in the membership coordinate for the full prefix $z$, so $f_A(x) \neq f_{A'}(x)$ and we are done. Thus assume

$$A(z) = 1 \quad \text{and} \quad A'(z) = 0,$$

and let $q' = \delta_{A'}(q_0, z)$ be the state that the DFA $A'$ reaches after traversing the string $z$. We distinguish two possibilities for $q'$.

**Case (i):** $q' \neq q_{\text{dead}}$. Because $A'$ is minimal, any non-accepting state distinct from the dead state has an accepting continuation (otherwise it would be equivalent to $q_{\text{dead}}$ and hence identified with it by minimality). Hence there exists a suffix $s \in \Sigma^*$ such that $A'(z \cdot s) = 1$.

Let $x := z \cdot s$. In the NSP label sequence for $x$, the membership coordinate at the prefix $z$ is

$$L_A(z) = 1 \quad \text{and} \quad L_{A'}(z) = 0,$$

so $f_A(x) \neq f_{A'}(x)$. The oracle may therefore return this positive example $x$ together with $f_{A'}(x)$.

**Case (ii):** $q' = q_{\text{dead}}$. Write $z = w_1 \cdots w_N$ with $w_i \in \Sigma$ and set $z_{:i} := w_1 \cdots w_i$ for $0 \leq i \leq N$ (with $z_{:0} = \lambda$). There exists an index $i \in \{0, \dots, N-1\}$ such that

$$\delta_{A'}(q_0, z_{:i}) \neq q_{\text{dead}} \qquad \text{and} \qquad \delta_{A'}(q_0, z_{:i+1}) = q_{\text{dead}}. \tag{3}$$

$A'(z) = 0$ and the DFA $A'$ ends at $q_{\text{dead}}$ after traversing $z$, so select the first position at which $q_{\text{dead}}$ is entered. By minimality (as in Case (i)), the non-dead state $\delta_{A'}(q_0, z_{:i})$ admits an accepting continuation; hence there exists a suffix $s \in \Sigma^*$ with $A'(z_{:i} \cdot s) = 1$.

For the continuation bit at the prefix $z_{:i}$ and the next symbol $w_{i+1}$, (3) implies

$$\varphi_{A'}(z_{:i}, w_{i+1}) = 0.$$

On the other hand, since $A(z) = 1$ and $z = z_{:i} \, w_{i+1} \, (w_{i+2} \cdots w_N)$, there exists a suffix (namely $w_{i+2} \cdots w_N$) witnessing that

$$\varphi_A(z_{:i}, w_{i+1}) = 1.$$

Now set $x := z_{:i} \cdot s$. This $x$ is accepted by $A'$, and the two NSP labelings disagree at the continuation coordinate $(z_{:i}, w_{i+1})$:

$$\left\| f_A(x) - f_{A'}(x) \right\|_\infty = 1.$$

Thus $x$ is a valid counterexample for the oracle.

In either case, there exists a positive example $x$ (accepted by $A'$) with $f_A(x) \neq f_{A'}(x)$ and if no such counterexample exists, then $A = A'$. $\qquad\square$

## G  LEARNING WITH MEMBERSHIP AND GENERATIVE QUERIES

In this section, we will show that the class of DFAs is PAC-learnable in the NSP setting with membership queries and generative queries which provide NSP labelled positive examples conditioned on input strings. The setting is inspired by the information one can obtain via blackbox access to the next token probabilities of any language model. When the language model is a formal device such as a Probabilistic DFA (See App. C), then we will (approximately) learn the support of the language model. For practical neural language models, our algorithm will learn the truncated support of the language model as defined in Section 2.2.

We will restate certain definitions and preliminaries in more detail here for clarity.

**Problem and Assumptions.** Let LM be a language model which induces a distribution over strings $\mathcal{D}_{\text{LM}}$. The support of the distribution $\mathcal{D}_{\text{LM}}$ is a regular language $L_{A^\star}$ which is recognized by the DFA $A^\star$. We will assume that the expected length of strings generated by the language model is finite. Given blackbox access to such a language model, we would like to find $\hat{A}$ such that $\mathbb{E}_{x \sim \mathcal{D}_{\text{LM}}}[\| f_{\hat{A}}(x) - f_{A^\star}(x) \|_\infty] \leq \epsilon$ with high probability.

Recall that, in this setting, a learner has access to two types of queries: (i) Membership queries $\text{MQ}(x) \in \{0, 1\}$ which returns $A^\star(x)$, and (ii) Generative queries $\text{Gen}_{\mathcal{D}_{\text{LM}}}(\cdot)$ which takes an input string or prompt $x$ and generates a string $s$ along with NSP labels based on the distribution $\mathcal{D}_{\text{LM}}$ conditioned on the prompt $x$. We describe how these queries can be simulated by practical language models in Section G.3.

The next statement is the main result, which formalizes that with an extension of $\text{L}^\star$, we can obtain an algorithm that, with high probability, outputs an automaton $\hat{A}$ using membership and generative queries such that $\mathcal{L}_{\text{NSP}}(f_{\hat{A}}; f_{A^\star}, \mathcal{D}_{\text{LM}}) \leq \epsilon$.

**Theorem 5.2.** *Let $A^\star \in \mathrm{DFA}_n$ be any minimal DFA with at most $n$ states, and let $\mathcal{D}_{\mathrm{LM}}$ be a distribution over strings whose support is $L_{A^\star}$. There exists an algorithm with access to the membership query oracle $\mathrm{MQ}$ and generative query oracle $\mathrm{Gen}_{\mathcal{D}_{\mathrm{LM}}}$ producing NSP labeled examples in $L_{A^\star}$, that runs in time polynomial in $n, 1/\epsilon, 1/\delta$ and the length of the largest string produced by the generative query oracle, and with probability at least $1 - \delta$, outputs a DFA $\hat{A}$ such that,*

$$\mathbb{E}_{x \sim \mathcal{D}_{\mathrm{LM}}} [\|f_{\hat{A}}(x) - f_{A^\star}(x)\|_\infty] \leq \epsilon.$$

To prove Theorem 5.2, we will show that there exists a consistent learner which takes $m$ NSP labelled examples from $\mathcal{D}_{\mathrm{LM}}$ and, by using the generative and membership query oracle, always returns a hypothesis $\hat{A}$ that is consistent with all the $m$ examples in terms of NSP labels. Our learning algorithm will be based on Angluin's $\mathrm{L}^\star$ algorithm (Angluin, 1987). We will first discuss the preliminaries of the $\mathrm{L}^\star$ framework based on modern and minimal treatments (Mohri et al., 2018; Kearns & Vazirani, 1994; Colcombet et al., 2021).

### G.1 L-STAR PRELIMINARIES

Let $\hat{A}$ be a hypothesis DFA constructed by a learner and let $A^\star$ be the target DFA. We will refer to a string $x$ such that $\hat{A}(x) \neq A^\star(x)$ as a *counterexample* (typically provided by an Equivalence query oracle).

**Access and Test words.** In this setup, we will have a pair of sets $(Q, T)$ where $Q \subseteq \Sigma^*$ is a set of *access words*, and $T \subseteq \Sigma^*$ is a set of *test words*. Both $Q$ and $T$ are always nonempty. Intuitively, the set $Q$ will play the role of states, where it will have a string corresponding to every state of our DFA. The test words $T$ act as distinguishing strings for the access words $Q$. Both the sets $Q, T$ contain the empty string $\lambda$. We will now define the notion of $T$-equivalence.

**Definition G.1** ($T$-Equivalence). *For a non-empty set $T \subseteq \Sigma^*$ and a language $L$, two strings $u, v$ are $T$-equivalent: $u \equiv_T v$, if for every string $s \in T$, the string $u \cdot s \in L$ if and only if $v \cdot s \in L$.*

**Closed and Separable.** A pair $(Q, T)$ is *closed* if for every access word $q \in Q$ and every symbol $\sigma \in \Sigma$, there exists an access word $q' \in Q$ such that $q \cdot \sigma \equiv_T q'$. A pair $(Q, T)$ is defined to be *separable* if every two distinct access words $q, q' \in Q$ are not T-equivalent: $q \not\equiv_T q'$.

**Constructing a DFA.** Given a closed and separable pair $(Q, T)$ and access to membership queries $\mathrm{MQ}$, one can construct a DFA as follows.

- Set the $q_0 = \lambda \in Q$.
- For every $q \in Q$, add them to the set of final states $F$ if $\mathrm{MQ}(q) = 1$.
- For any access word $q$ and symbol $\sigma \in \Sigma$, let $q'$ be the word such that $q \cdot \sigma \equiv_T q'$. Then set the transition $\delta(q, \sigma) = q'$. Since $(Q, T)$ is closed, such a state or access word $q'$ must exist, and separability implies uniqueness of such a word.

To prove the correctness of our algorithm, we use the following technical lemmas from $\mathrm{L}^\star$ without proof.

**Lemma G.1** ($\mathrm{L}^\star$ Fact (Angluin, 1987)). *Let $(Q, T)$ be a closed and separable pair of sets which is consistent with some language $L$. Let $A^\star$ be the minimal automaton that decides $L$. Then, $|Q| \leq |A^\star|$.*

The above Lemma follows directly from Lemma C.1 and the fact that we have distinguishing string in $T$ for each pair of strings in $Q$.

**Lemma G.2** ($\mathrm{L}^\star$ Fact (Angluin, 1987)). *Let $(Q, T)$ be a pair that is separable but not closed. Then, using at most $|Q||T|(|\Sigma| + 1)$ membership queries, and in time polynomial in $(|Q|, |T|, |\Sigma|)$, we can identify $q' \notin Q$, such that $(Q \cup \{q'\}, T)$ is separable.*

The following lemma is the crux of the original $\mathrm{L}^\star$ algorithm in terms of how it updates the pair $(Q, T)$ based on disagreement between the hypothesis DFA and a labelled example.

**Lemma G.3** (L$^\star$ Fact (Angluin, 1987)). *Let $(Q, T)$ be a closed and separable pair, and $\hat{A}$ be the associated DFA. Suppose $x \in \Sigma^*$ be a string such that $\hat{A}(x) \neq A^\star(x)$. Then using at most $|x|$ membership queries and in time polynomial in $|x|$, we can identify $q' \notin Q$ and $t' \notin T$ such that $(Q \cup \{q'\}, T \cup \{t'\})$ is separable.*

### G.2 PROCESSING COUNTEREXAMPLES IN THE NSP SETTING

Similar to L$^\star$, our L$^\star_{\text{nsp}}$ algorithm will maintain a pair $(Q, T)$ and iteratively expand the sets by finding new states and distinguishing strings based on disagreements with the training examples. However, in the NSP setting, the disagreements in labels need not always be based on the differences in the memberships of strings, but can be based on the continuation labels. Hence, Lemma G.3 does not directly yield us our desired algorithm. The key difference between L$^\star$ and L$^\star_{\text{nsp}}$ is in how they obtain and process disagreements between predicted labels and true NSP labels.

Let $\hat{A}$ be the hypothesis DFA constructed by a learner based on a closed and separable $(Q, T)$ and $A^\star$ be the target DFA. In the following lemma, we will show that when presented with an NSP labelled $x$ such that $f_{\hat{A}}(x) \neq f_{A^\star}(x)$ (either from training set or an Equivalence query oracle), the learner can find at least one new pair of access word $q' \notin Q$ and test word $t' \notin T$ using membership and generative queries.

**Lemma 5.1.** *Let $(Q, T)$ be closed and separable, and let $\hat{A}$ be the minimal DFA induced by $(Q, T)$. Suppose there exists a string $x$ with $f_{\hat{A}}(x) \neq f_{A^\star}(x)$. Then one can find $q' \notin Q$ and $t' \notin T$ such that $(Q \cup \{q'\}, T \cup \{t'\})$ is separable, using membership queries and at most one generative query in polynomial time. If a generative query is used, let $y$ denote its output; otherwise set $y = \lambda$. The total number of membership queries is at most $|x| + |y|$, and the running time is polynomial in $|x| + |y| + |Q|$.*

*Proof.* By definition, we have a string such that $A^\star(x) = 1$ and $f_{\hat{A}}(x) \neq f_{A^\star}(x)$. Thus, there is a prefix $x_{:n}$ of $x$ at which the NSP labels disagree; the disagreement is either (a) in the *membership* label for $x_{:n}$, or (b) in one of the *continuation* labels for some symbol $\sigma \in \Sigma$. We treat these two cases separately. For all cases, our goal will be to construct a string $x'$ such that we can invoke Lemma G.3 to obtain $q'$ and $t'$.

**Case A:** Membership-label disagreement. There exists a prefix $x_{:n}$ of $x$ such that $A^\star(x_{:n}) \neq \hat{A}(x_{:n})$. In this case we may simply take $x' = x_{:n}$ and apply Lemma G.3. That lemma produces an access word $q' \notin Q$ and a test word $t' \notin T$ in time polynomial in $|x'|$ (and using at most $|x'|$ membership queries), and it guarantees that $(Q \cup \{q'\}, T \cup \{t'\})$ is separable.

**Case B:** Continuation-label disagreement. There exists a prefix $x_{:n}$ of $x$ and a symbol $\sigma \in \Sigma$ such that

$$\varphi_{\hat{A}}(x_{:n}, \sigma) \neq \varphi_{A^\star}(x_{:n}, \sigma).$$

We distinguish two subcases according to the value of the target continuation label.

*Subcase B1:* $\varphi_{A^\star}(x_{:n}, \sigma) = 0$ and $\varphi_{\hat{A}}(x_{:n}, \sigma) = 1$. By the semantics of the continuation labels for the target, $\varphi_{A^\star}(x_{:n}, \sigma) = 0$ means that for every suffix $s \in \Sigma^*$ we have $A^\star(x_{:n} \cdot \sigma \cdot s) = 0$. On the other hand, $\varphi_{\hat{A}}(x_{:n}, \sigma) = 1$ means that the state $\tilde{q} := \delta_{\hat{A}}(q_0, x_{:n} \cdot \sigma)$ that $\hat{A}$ reaches after traversing $x_{:n} \cdot \sigma$ is not the dead state.

Therefore, we can search within $\hat{A}$ from $\tilde{q}$ to see whether there is some accepting continuation. Run a breadth–first search in $\hat{A}$ starting at $\tilde{q}$; if the search reaches a final state $q \in F_{\hat{A}}$, let $s'$ be a shortest suffix labeling such a path. Then by construction, the length $|s'| \leq |Q|$ and

$$A^\star(x_{:n} \cdot \sigma \cdot s') = 0 \qquad \text{and} \qquad \hat{A}(x_{:n} \cdot \sigma \cdot s') = 1,$$

so $x' := x_{:n} \cdot \sigma \cdot s'$ is a standard membership counterexample. Applying Lemma G.3 to $x'$ yields $q' \notin Q$ and $t' \notin T$ in polynomial time.

*Subcase B2:* $\varphi_{A^\star}(x_{:n}, \sigma) = 1$ and $\varphi_{\hat{A}}(x_{:n}, \sigma) = 0$. The target label $\varphi_{A^\star}(x_{:n}, \sigma) = 1$ asserts that there exists a suffix $s' \in \Sigma^*$ with $A^\star(x_{:n} \cdot \sigma \cdot s') = 1$. In contrast, $\varphi_{\hat{A}}(x_{:n}, \sigma) = 0$ means that $\delta_{\hat{A}}(\delta_{\hat{A}}(q_0, x_{:n}), \sigma) = q_{\text{dead}}$, hence $\hat{A}(x_{:n} \cdot \sigma \cdot s) = 0$ for every suffix $s$.

As a consequence of Prop. E.1, we also have that such a suffix cannot be found in polynomial time using membership queries only. Thus, to find an accepting suffix, we call the generative query oracle with input $x_{:n} \cdot \sigma$. Since $\varphi_{A^\star}(x_{:n}, \sigma) = 1$, such a suffix is guaranteed to exist. Let $s' = \text{Gen}_{\mathcal{D}_{\text{LM}}}(x_{:n} \cdot \sigma)$ and we have $x' = x_{:n} \cdot \sigma \cdot s'$ which is rejected by $\hat{A}$ (since it reaches a dead state after reading $x_{:n} \cdot \sigma$). Thus, invoking Lemma G.3 on $x'$ produces the desired $q'$ and $t'$ in polynomial time.

In all cases we obtain $q' \notin Q$ and $t' \notin T$ such that $(Q \cup \{q'\}, T \cup \{t'\})$ is separable. This completes the proof. $\qquad\square$

**Theorem G.4.** *Let $A^\star$ be the minimal DFA that recognizes the support language $L_{A^\star} \subseteq \Sigma^*$ of the distribution $\mathcal{D}_{\text{LM}}$. Given $m$ NSP labelled examples $X = \langle x^{(i)}, f_{A^\star}(x^{(i)}) \rangle_{i=1}^m$, with access to membership query oracle $\text{MQ}$ and generative query oracle $\text{Gen}_{\mathcal{D}_{\text{LM}}}$ with respect to $\mathcal{D}_{\text{LM}}$, Algorithm 1 outputs a DFA $\hat{A}$ such that $f_{\hat{A}}(x^{(i)}) = f_{A^\star}(x^{(i)})$ for all $i = 1, \dots, m$. The running time is polynomial in $|A^\star|, |\Sigma|, \max_i |x^{(i)}|$, and the length of the longest string returned by the generative query oracle.*

*Proof.* If the target language is $L_{A^\star} = \emptyset$, then the initial hypothesis DFA $\hat{A}$ at Line 6 will also be such that $\hat{A}(x) = 0$ for all $x \in \Sigma^*$. Whenever a disagreement with any training example is found, by Lemma 5.1, the algorithm will add at least string to $Q$ and effectively identify at least one new state from $A^\star$. By Lemma G.1, we have that $|Q| \leq |A^\star|$ and thus after at most $|A^\star|$ iterations of the main loop (Line 7) or equivalently, at most $|A^\star|$ calls to Lemma G.3, we have that $|Q| = |A^\star|$. Each state in $Q$ is uniquely identified with a set of distinguishing strings that are consistent with $A^\star$. Since $(Q, T)$ is closed and the transitions are defined using membership queries to check $T$-equivalence, the final DFA is isomorphic to the target DFA $A^\star$. Hence, in such a case $\hat{A} = A^\star$ and the hypothesis must be consistent with all training examples. Thus, it will terminate. The algorithm can of course terminate with $|Q| < |A^\star|$ if the hypothesis DFA is consistent with all the training examples.

$\qquad\square$

As a consequence of the above theorem and the fact that $\log |\text{DFA}_n| = O(n \log n)$ (Lemma C.3), Theorem 5.2 follows from standard Occam's razor arguments.

### G.3 SIMULATING MEMBERSHIP AND GENERATIVE QUERIES

We describe how membership and generative queries can be conveniently simulated by blackbox access to the target language model LM. For a neural language model, the empty string corresponds to [BOS] token.

**(i)** *Membership Queries.* Given a query string $x = w_1 \cdots w_N \in \Sigma^*$, compute the NSP continuation labels for every prefix of $x$ using the language model including the empty string $\lambda$. For a neural language model, this is simply done by obtain the next token probabilities for each prefix and applying the truncation rule $\mathcal{T}$ (cf. Sec. 2.2). If for any prefix the continuation label $\varphi(x_{:n}, w_{n+1}) = 0$ then $w_{n+1}$ is not a valid continuation so the oracle can return 0. If the path traversed by $w_1, \cdots, w_N$ is valid, then we can check whether $\varphi(x, [\text{EOS}])$ is 1 or 0 and thus $\text{MQ}(x) = \varphi(x, [\text{EOS}])$.

**(ii)** *Generative Queries.* For any string or prefix $x$, the Generative Query Oracle $\text{Gen}_{\mathcal{D}_{\text{LM}}}(x)$ provides $(s, \boldsymbol{y})$ where $s$ is a string from the distribution $\mathcal{D}_{\text{LM}}^{\mathcal{T}}$ conditioned $x$. If no such continuation exists and $x$ is at a dead state, then it returns 'None'. To simulate the example oracle or equivalently, to obtain examples from the target distribution $\mathcal{D}_{\text{LM}}^{\mathcal{T}}$, one can simply query the oracle with empty string $\text{Gen}_{\mathcal{D}_{\text{LM}}}(\lambda)$. To obtain a continuation for a particular prefix $x$, one can provide it as an input prompt to the language model and first check whether $x$ traverses a valid path based on the NSP labels as described above. If it does not then return 'None'. If it does then, then iteratively sample the next tokens using the truncation rule until the [EOS] is permissible $\varphi(x \cdot s, [\text{EOS}]) = 1$.

### G.4 ON EXACT LEARNING WITH MEMBERSHIP AND EQUIVALENCE QUERIES

We show that the class of DFAs can be learned exactly with membership queries and *two types* of equivalence queries. (i) $\text{EQ}_{\text{nsp}}^+(A; A^\star)$ which returns a positive NSP-labelled counterexample with

---

**Algorithm 1** $\mathrm{L}^{\star}_{\mathrm{nsp}}$ algorithm

---

1: $Q \leftarrow \{\lambda\}, \quad T \leftarrow \{\lambda\}$
2: $X = \langle x^{(i)}, f_{A^{\star}}(x^{(i)})\rangle_{i=1}^{m}$          $\triangleright$ call $\mathrm{Gen}_{\mathcal{D}_{\mathrm{LM}}}(\lambda)$ $m$ times
3: **while** $(Q, T)$ not closed **do**
4:     Use Lemma G.2 to obtain $q'$;  $Q \leftarrow Q \cup \{q'\}$
5: **end while**
6: Construct hypothesis automaton $\hat{A}$ from $(Q, T)$
7: **while** $\hat{A}$ not consistent with $X$ **do**
8:     Choose $x \in X$ with $f_{\hat{A}}(x) \neq f_{A^{\star}}(x)$
9:     **while** $f_{\hat{A}}(x) \neq f_{A^{\star}}(x)$ **do**          $\triangleright$ Use Lemma 5.1
10:         Let $x_{:n}$ be the prefix with first NSP mismatch
11:         let $\sigma$ denote the symbol if it is a continuation mismatch
12:         **if** membership mismatch at $x_{:n}$   **then**          $\triangleright$ Case A
13:             $w' \leftarrow x_{:n}$
14:         **else if** continuation mismatch with $A^{\star}$ forbids and $\hat{A}$ admits   **then**          $\triangleright$ Case B1
15:             $s' \leftarrow$ find $\hat{A}$-accepting suffix from $x_{:n} \cdot \sigma$
16:             $w' \leftarrow x_{:n} \cdot \sigma \cdot s'$
17:         **else**          $\triangleright$ Case B2
18:             $w' \leftarrow x_{:n} \cdot \sigma \cdot \mathrm{Gen}_{\mathcal{D}_{\mathrm{LM}}}(x_{:n} \cdot \sigma)$
19:         **end if**
20:         Use Lemma G.3 on $w'$ to obtain $q'$ and $t'$
21:         $Q \leftarrow Q \cup \{q'\}, \; T \leftarrow T \cup \{t'\}$
22:         **while** $(Q, T)$ not closed **do**
23:             Use Lemma G.2 to obtain $q'$;  $Q \leftarrow Q \cup \{q'\}$
24:         **end while**
25:         Reconstruct $\hat{A}$ from $(Q, T)$
26:     **end while**
27: **end while**

---

at least one disagreement in NSP labels or else it returns equivalence; (ii) $\mathrm{EQ}_{\mathrm{pos}}(A; A^{\star})$ which just returns a string with no additional labels, such that the string is accepted by $A^{\star}$ but rejected by $A$ if there exists such a string, otherwise it returns none. The problem of learning with just membership and NSP equivalence queries remains open.

For the setting where a learner has access to two types of equivalence queries, the algorithm is almost the same as the one described earlier (Alg. 1) with a couple of differences. Similar to the previous algorithm with membership and generative queries, the learner will maintain a pair $(Q, T)$ and iteratively update them, but with counterexamples from $\mathrm{EQ}_{\mathrm{nsp}}$ instead of a set of labelled examples. There are only two changes to address, which we describe below.

**(i)** The first thing to note is that by Proposition 3.1, the equivalence query oracle $\mathrm{EQ}^{+}_{\mathrm{nsp}}$ is well-defined and always guaranteed to return a counterexample. If the target DFA $A^{\star}$ is such that $L_{A^{\star}} = \emptyset$, then the initial hypothesis DFA in Line 6 of Alg. 1 will be the same as $A^{\star}$ and thus, we do not need any counterexample.

**(ii)** Secondly, we do not have access to a generative query oracle anymore, so we cannot use it to find an accepting continuation when the target NSP label says that a suffix exists, but for our hypothesis DFA, no such suffix exists. The difficulty is that $\mathrm{EQ}^{+}_{\mathrm{nsp}}$ against our hypothesis need not return a counterexample beginning with the particular prefix $x_{:n} \cdot \sigma$. To force such a witness, we will use $\mathrm{EQ}_{\mathrm{pos}}(A, A^{\star})$. Consider the DFA $A_{x_{:n}\sigma}$ whose language is

$$L_{A_{x_{:n}\sigma}} := \Sigma^{*} \setminus (x_{:n} \cdot \sigma \cdot \Sigma^{*}),$$

i.e., $A_{x_{:n}\sigma}$ accepts exactly the strings that do *not* begin with the prefix $x_{:n} \cdot \sigma$. Submit the equivalence query $\mathrm{EQ}_{\mathrm{pos}}(A_{x_{:n}\sigma}; A^{\star})$. Because $\varphi_{A^{\star}}(x_{:n}, \sigma) = 1$, there exists an accepted string with prefix $x_{:n} \cdot \sigma$; therefore, the oracle must return a positive counterexample of the form

$$x' = x_{:n} \cdot \sigma \cdot s' \quad \text{with} \quad A^{\star}(x') = 1.$$

But $x'$ is rejected by $\hat{A}$ (the run reaches the dead state after reading $x_{:n} \cdot \sigma$), so $x'$ is again a standard membership counterexample.

Figure 5: Key metrics for $L^\star_{\text{nsp}}$ on other Tomita grammars. See Sec. 6 for details.

**Correction Queries.** In the above approach and Sec. 5, we use generative queries and the positive equivalence query oracle to find an accepting suffix. Becerra-Bonache et al. (2006) proposed a learning model where the learner has access to something called correction queries (CQ). For a string $x \in \Sigma^*$, using the query $\text{CQ}(x)$ will return the shortest accepting suffix (i) if $x$ is not in the target language and (ii) if such an accepting suffix exists. If the $x$ is in the target language, then the oracle can return an empty string, and if no accepting suffix exists, then CQ returns a special symbol not in the alphabet $\Sigma$. Thus, CQ provides the information provided by an MQ along with an accepting suffix when it exists. Becerra-Bonache et al. (2006) analyze learnability with correction and equivalence query oracle in the conventional classification setting. CQ and generative queries are closely related in the sense that generative queries can be seen as a distribution-dependent version of CQ along with additional NSP labels. In the NSP setting, using the approaches discussed above, it is straightforward to see that the class of DFAs is exactly learnable from positive examples only with a correction query CQ and equivalence query oracle $\text{EQ}^+_{\text{nsp}}$.

It is unclear whether exact learning with just membership and the NSP equivalence queries is feasible and is left as an open problem. When the learner gets a counterexample such that the only disagreement is of the form that $\varphi_{A^\star}(x_{:n}, \sigma) = 1$ and $\varphi_{\hat{A}}(x_{:n}, \sigma) = 0$, then one might wonder if a continuation can be recovered using the NSP membership query oracle $\text{MQ}_{\text{nsp}}$ which provides additional label. But as a consequence of Prop. E.1, such an accepting continuation cannot be found in polynomial time using $\text{MQ}_{\text{nsp}}$.

Table 1: Tomita grammars and descriptions.

| Tomita grammar | Description |
| --- | --- |
| Tomita 1 | Strings of only 1s (including the empty string), $1^*$. |
| Tomita 2 | Repeated "10" pattern, $(10)^*$. |
| Tomita 3 | Any block with an *odd* number of consecutive 1s is always followed by a block with an *even* number of consecutive 0s. |
| Tomita 4 | Strings that do not contain `000` as a substring. |
| Tomita 5 | Strings with an even number of 0s and an even number of 1s. |
| Tomita 6 | Strings where the difference between the number of 0s and the number of 1s is a multiple of 3. |
| Tomita 7 | $0^*1^*0^*1^*$ (zero or more 0s, then 1s, then 0s, then 1s). |

# H  FURTHER EXPERIMENTS AND DETAILS

We discuss additional details of the experimental setup as well as additional results for DFA extraction from language models.

## H.1  ADDITIONAL DETAILS OF THE SETUP

**Tasks.** We provide additional details about the tasks considered in our experiments.

*Tomita grammars.* Table 1 provides the description of the 7 Tomita grammars (Tomita, 1982) in the benchmark. We use all of them except the first one since it is trivial ($1^*$), and the results are uninfor-

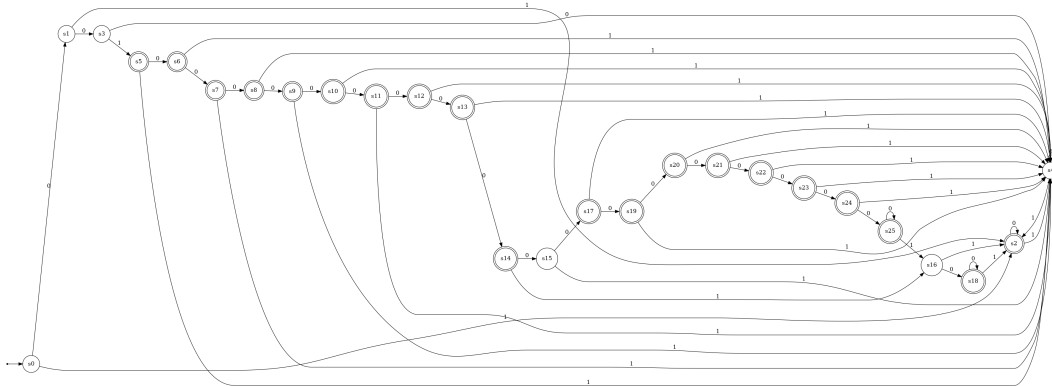

Figure 6: DFA with 26 states extracted by $L^\star_{\mathrm{nsp}}$ from Transformer trained on Parity. See App. H.2 for more details.

mative. All the regular languages in the Tomita benchmark have 2-5 states. These benchmarks have primarily been used to extract DFAs from RNN or Transformer-based classifiers (Wang et al., 2018; Weiss et al., 2018; Zhang et al., 2024).

*Parity.* The Parity language is a simple two-state DFA recognizing whether the number of 1s in a string is odd or even. This task has been widely in the context of Transformers and previous works have shown empirical (Bhattamishra et al., 2020a) and theoretical (Hahn & Rofin, 2024; Hahn, 2020) evidence to indicate that such functions are difficult for Transformers to model.

*Bounded Dycks.* DYCK-$(n, k)$ represents the language with well-balanced parentheses with $n$ types of brackets and depth at most $k$. These languages have also been widely used in analysis of sequence models (Hewitt et al., 2020; Bhattamishra et al., 2020b) since they capture hierarchical dependencies prevalent in natural languages. These DFAs have a relatively larger number of states than Tomita grammars and Parity, and hence provide a testbed to evaluate learning algorithms on languages with increasing state complexity. We use the following languages in our experiments: DYCK-$(2, 2)$: 8 states, DYCK-$(2, 4)$: 32 states, DYCK-$(3, 3)$: 41 states, and DYCK-$(4, 3)$: 86 states.

**Compute.** The experiments in the paper are not compute heavy. All our experiments were conducted using 8 NVIDIA Tesla V100 GPUs each with 16GB memory for training language models. Each run for up to 40k steps could take 1-6 hours, depending on the task. Some runs are much shorter if the model achieves high accuracy quite early. The execution of $L^\star_{\mathrm{nsp}}$ is primarily on CPU, and uses the GPU to answer Generative and Membership queries with the Transformer language model. The time taken for each run of $L^\star_{\mathrm{nsp}}$ is provided in the main plots (Fig. 2 and 5).

## H.2 IDENTIFYING ERRONEOUS EXAMPLES

When the Transformer models trained on regular languages are not perfect, the extracted DFA is not isomorphic to the target DFA used to generate training data for Transformers.

There are two key things to note about this event. (i) When the extracted DFA has more states than the target DFA (such as for Parity in Fig. 2), then it implies that the support language for Transformer has at least as many states as the extracted DFA. This is because states are always created after identifying distinguishing strings (Lemma C.1) using the teacher model. (ii) The cases where models are not perfectly trained and our extracted DFA identifies erroneous strings, the models are still well-trained in the sense that all 1k strings generated using the language model are in the target language. Thus, the difference comes from disagreement in some NSP label which the $L^\star_{\mathrm{nsp}}$ algorithm then leverages to extract a bigger DFA, which is then used to identify erroneous strings.

**Method.** When the learned DFA $\hat{A}$ is not isomorphic to the target DFA $A^\star$, we construct the product automaton $B = \hat{A} \times A^\star$ where each state of $B$ is a pair $(p, q)$ with $p \in Q(\hat{A})$ and $q \in Q(A^\star)$; on input symbol $a$, $B$ transitions $\delta_B((p, q), a) = (\delta_{\hat{A}}(p, a), \delta_{A^\star}(q, a))$. The state $(p, q)$ is accepting if and only if exactly one of $p$ or $q$ is accepting (XOR). The product DFA accepts the strings in the symmetric difference of the hypothesis and target DFA $L(B) = L(\hat{A}) \triangle L(A^\star)$. We then enumerate

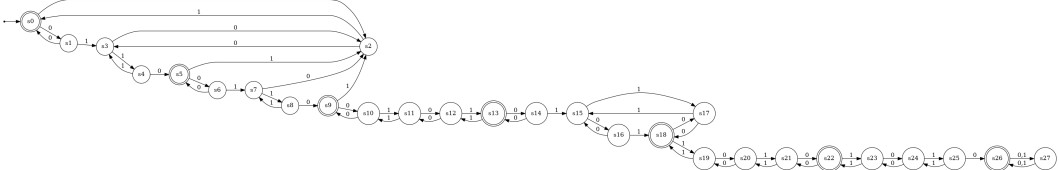

Figure 7: DFA with 28 states extracted by $L^\star_{nsp}$ from Transformer trained on Tomita-5. See App. H.2 for more details.

erroneous strings by doing a BFS-like search within $B$ to find accepting strings in nondecreasing length order.

Table 2: Examples of erroneous strings found via extracted DFAs. The $A^\star$ column within states denotes the number of states in the target DFA used to train the language model. The $\hat{A}$ column indicates the number of states in the extracted DFA. Ground truth labels are denoted by $y_\star$, and Teacher LM labels by $y_T$. Note: The models used as Teacher LM for Tomita-5, DYCK-$(2,2)$ and DYCK-$(3,3)$ for these results were imperfect and not the same models used for the experiments in Figure 2. See Sections 6 and H.2 for more details.

| Language | States | | Labels | | Erroneous string $x$ |
|---|---|---|---|---|---|
| | $|A^\star|$ | $|\hat{A}|$ | $y_\star$ | $y_T$ | |
| Parity | 2 | 26 | 1 | 0 | 0 0 1 0 0 0 0 0 0 0 0 0 0 0 |
| | | | 0 | 1 | 0 0 1 0 0 0 0 0 0 0 0 0 0 1 0 |
| | | | 0 | 1 | 0 0 1 0 0 0 0 0 0 0 0 0 1 0 0 |
| Dyck-(2,2) | 8 | 189 | 0 | 1 | ( ) [ ] ( ) ( ) ( ) [ ] ( ) [ ] ] |
| | | | 0 | 1 | [ ] [ ] ( ) ( ) ( ) [ ] ( ) [ ] ] |
| | | | 0 | 1 | ( ) [ ] ( ) ( ) ( ) [ ] ( ) [ ] ] ( ) |
| Dyck-(3,3) | 41 | 87 | 0 | 1 | { { ( ) [ ] [ ] [ ] } ( ) { } |
| | | | 0 | 1 | { { [ ] [ ] [ ] [ ] } ( ) { } |
| | | | 0 | 1 | { { { } [ ] [ ] [ ] } ( ) { } |
| Tomita-5 | 4 | 28 | 0 | 1 | 0 1 1 0 0 1 1 0 0 1 0 1 0 1 0 1 1 0 1 0 1 0 1 0 0 1 |
| | | | 0 | 1 | 0 1 1 0 0 1 1 0 0 1 0 1 0 1 0 1 1 0 1 0 1 0 1 0 1 0 |
| | | | 0 | 1 | 0 1 1 0 0 1 1 0 0 1 0 1 0 1 1 0 1 0 1 0 1 0 1 0 0 1 |

**Results.** We observed erroneous strings for languages like Parity, Tomita-5, DYCK-$(2,2)$, and DYCK-$(3,3)$. Examples of some erroneous strings identified by the hypothesis DFA is provided in Table 2. Figure 6 and 7 show the DFAs extracted for Parity and Tomita-5, respectively. The DFAs for DYCK-$(2,2)$ and DYCK-$(3,3)$ are too large to be visually interpretable. Constructing the product DFA is efficient and identifying several erroneous examples takes only a few seconds. There is no natural distribution over the symmetric difference language and further it can even be finite in some cases which makes it difficult to systematically compute the accuracy of predicting erroneous examples using the extracted DFA. The closest signal we have is the NSP accuracy for the extracted DFAs which is near perfect.

Fortunately, identifying strings in the symmetric difference language of $\hat{A}$ and $A^\star$ is quite efficient which can allow one to find numerous erroneous examples (if they exist). For instance, with 1k strings generated from Transformer trained on Parity, the $L^\star_{nsp}$ algorithm extracted a DFA with 26 states within 10 seconds, and using the extracted DFA, we identified 1k strings in the symmetric difference language $L(\hat{A}) \triangle L(A^\star)$ in 16 seconds. Out of the 1000 strings identified by $\hat{A}$, 987 of them were erroneous: $A^\star(x) \neq \mathrm{MQ}(x)$ where $\mathrm{MQ}(x)$ denotes whether the string was in the support of the teacher language model and $A^\star(x)$ indicates whether the string was in the target regular language. Thus, such an approach can sometimes be far more efficient than finding erroneous strings by repeatedly sampling from a language model and testing whether they are correct or not.

For all languages except Parity, the language model became more robust when we retrained them for longer steps. The curves for Tomita-5, and the two Dyck languages in Fig. 2 are with retrained

and more accurate Transformer models. We suspect that for the lengths considered in this paper, retraining a larger Transformer on Parity for longer might make it more robust, but we keep the imperfect model to illustrate the differences in Fig. 2.

### H.3    PRACTICAL NOTES ON THE ALGORITHM

As mentioned in Sec. 7, if the DFA representing the support of the language model has a large number of states, then the extraction algorithm can be extremely slow. The key bottleneck for $L^\star_{nsp}$ is the same as $L^\star$, which is the closure step (Line 22 in Alg. 1). Even though it is polynomial, the time complexity $O(|Q||T||\Sigma|)$ blows up when both $|Q|$ and $|T|$ are in the order of thousands, since the process is sequential. We observe that when the models are partially or poorly trained, then the number of states blows up, and the algorithm slows down significantly, failing to terminate even after a day.

The other step where the $L^\star_{nsp}$ algorithm might fail is if the assumptions about the learning problem are violated in which case the use of generative queries to find accepting suffixes (Line 18 in Alg. 1) may not terminate. If the support language is not regular, then even though a continuation is permissible according to min-p/top-p sampling, the language model may still not terminate. However, we did not observe any instance of this problem in all our experiments. We suspect that since the language models are typically trained to predict [EOS] after a certain steps, they always terminate (EOS token becomes permissible) after a certain number of inference steps.

**Tokenization.** Another practical consideration concerns the effect of tokenizers. The algorithm can be applied to character-level language models and to word-level models with distinct word vocabularies in a natural way, since in both cases the alphabet aligns closely with the underlying notion of a string. For subword or BPE tokenizers, while the algorithm can be applied as is, one needs to be more careful about the semantics of the extracted DFA. The algorithm treats the language model as a black-box sequence model over an alphabet $\Sigma$ of discrete symbols, so in all cases the learned DFA captures the model's behavior over sequences of elements of $\Sigma$, and our guarantees apply at this level. However, with subword tokenizers there is typically no one-to-one correspondence between token sequences and character strings, since a single surface string may have multiple tokenizations that the model treats differently. As a consequence, the DFA learned over tokens should be interpreted as describing the token-level support induced by the fixed tokenizer and decoding rule, and cannot, in general, be read directly as an automaton for a specified character-level regular language.

**Efficiency improvements.** While the algorithm is polynomial-time, it could still be challenging to run it at scale as described above. To mitigate those, there are few natural ways to avoid redundant computation and improve efficiency in practice. *(i) LRU Cache for MQ.* In our implementation, we use an LRU cache with the membership query (MQ) oracle. The key computational bottleneck in the algorithm is that it requires many MQ calls in the closure steps where each MQ call is essentially a forward pass through the teacher language model. To avoid querying the label of the same string multiple times, we have an LRU cache which stores the last $10^5$ recently used strings which when queried directly return the label instead of making a pass through the teacher LM. *(ii) Parallelization.* Another concrete direction to improve efficiency is to parallelize the calls in the closure steps. The key observation is that the MQ calls in the closure step need not be in any particular order. Hence, one could query the labels of multiple strings by processing them as a batch through the teacher LM. To do so along with the LRU cache, one could take the following approach: suppose the closure steps requires $K$ MQ calls and let batch size be $B$. One could sequentially go through the strings to be queried and add them to a queue if they are not in the LRU cache and if a string is in the cache, then directly return its label. If the size of the queue reaches $B$ or there are no more strings to be queried, then algorithm can pass the items in the queue as a batch through the teacher LM. This could lead to tangible gains with GPUs which can fit large batches.

### H.4    ABLATION: USAGE OF CONTINUATION LABELS

When we apply the $L^\star_{nsp}$ algorithm on positive examples, information about refinements or new states is provided by two types of labels: (i) prefix membership labels, and (ii) continuation labels. Within continuation labels, there are two cases as described in Lemma 5.1. Whenever the algorithm finds one of the three types of disagreements in the NSP labels between the hypothesis DFA and the examples in its set, it refines the DFA and adds one more state to its hypothesis.

Table 3: Usage of different types of NSP labels by the $L_{nsp}^{\star}$ algorithm on various tasks. Refinement indicates the number of times states were added through disagreements (number of times loop in Line 9 is executed or equivalently Lemma 5.1 is invoked. Mem fraction denotes the percentage of times prefix membership labels were used. B1 and B2 fractions denote how many continuation labels were used. See Sec. H.4 for more details.

| Language | States | Prefix Memberships (%) | Cont. B1 (%) | Cont. B2 (%) | Refinements |
|---|---|---|---|---|---|
| **Sample size: 10** | | | | | |
| Dyck22 | $8.0 \pm 0.0$ | $0.0 \pm 0.0$ | $16.7 \pm 0.0$ | $\mathbf{83.3 \pm 0.0}$ | $6.00 \pm 0.00$ |
| Dyck24 | $21.0 \pm 5.4$ | $0.0 \pm 0.0$ | $5.9 \pm 2.6$ | $\mathbf{94.1 \pm 2.6}$ | $19.00 \pm 5.35$ |
| Dyck33 | $32.5 \pm 7.0$ | $0.0 \pm 0.0$ | $3.5 \pm 1.2$ | $\mathbf{96.5 \pm 1.2}$ | $30.50 \pm 7.01$ |
| Dyck43 | $37.4 \pm 14.1$ | $0.0 \pm 0.0$ | $3.8 \pm 3.2$ | $\mathbf{96.2 \pm 3.2}$ | $35.40 \pm 14.14$ |
| Parity | $2.0 \pm 0.0$ | $\mathbf{0.0 \pm 0.0}$ | $\mathbf{0.0 \pm 0.0}$ | $\mathbf{0.0 \pm 0.0}$ | $0.00 \pm 0.00$ |
| Tomita 2 | $3.0 \pm 0.0$ | $0.0 \pm 0.0$ | $0.0 \pm 0.0$ | $100.0 \pm 0.0$ | $1.00 \pm 0.00$ |
| Tomita 5 | $4.0 \pm 0.0$ | $\mathbf{100.0 \pm 0.0}$ | $0.0 \pm 0.0$ | $0.0 \pm 0.0$ | $2.00 \pm 0.00$ |
| Tomita 7 | $5.0 \pm 0.0$ | $0.0 \pm 0.0$ | $\mathbf{100.0 \pm 0.0}$ | $0.0 \pm 0.0$ | $3.00 \pm 0.00$ |
| **Sample size: 100** | | | | | |
| Dyck22 | $8.0 \pm 0.0$ | $0.0 \pm 0.0$ | $16.7 \pm 0.0$ | $\mathbf{83.3 \pm 0.0}$ | $6.00 \pm 0.00$ |
| Dyck24 | $32.0 \pm 0.0$ | $0.0 \pm 0.0$ | $3.3 \pm 0.0$ | $\mathbf{96.7 \pm 0.0}$ | $30.00 \pm 0.00$ |
| Dyck33 | $41.0 \pm 0.0$ | $0.0 \pm 0.0$ | $2.6 \pm 0.0$ | $\mathbf{97.4 \pm 0.0}$ | $38.90 \pm 0.32$ |
| Dyck43 | $85.6 \pm 1.3$ | $0.0 \pm 0.0$ | $1.2 \pm 0.0$ | $\mathbf{98.8 \pm 0.0}$ | $83.60 \pm 1.26$ |
| Parity | $2.0 \pm 0.0$ | $\mathbf{0.0 \pm 0.0}$ | $\mathbf{0.0 \pm 0.0}$ | $\mathbf{0.0 \pm 0.0}$ | $0.00 \pm 0.00$ |
| Tomita 2 | $3.0 \pm 0.0$ | $0.0 \pm 0.0$ | $0.0 \pm 0.0$ | $100.0 \pm 0.0$ | $1.00 \pm 0.00$ |
| Tomita 5 | $4.0 \pm 0.0$ | $\mathbf{100.0 \pm 0.0}$ | $0.0 \pm 0.0$ | $0.0 \pm 0.0$ | $2.00 \pm 0.00$ |
| Tomita 7 | $5.0 \pm 0.0$ | $0.0 \pm 0.0$ | $\mathbf{100.0 \pm 0.0}$ | $0.0 \pm 0.0$ | $3.00 \pm 0.00$ |
| **Sample size: 1000** | | | | | |
| Dyck22 | $8.0 \pm 0.0$ | $0.0 \pm 0.0$ | $16.7 \pm 0.0$ | $\mathbf{83.3 \pm 0.0}$ | $6.00 \pm 0.00$ |
| Dyck24 | $32.0 \pm 0.0$ | $0.0 \pm 0.0$ | $3.3 \pm 0.0$ | $\mathbf{96.7 \pm 0.0}$ | $30.00 \pm 0.00$ |
| Dyck33 | $41.0 \pm 0.0$ | $0.0 \pm 0.0$ | $2.6 \pm 0.0$ | $\mathbf{97.4 \pm 0.0}$ | $39.00 \pm 0.00$ |
| Dyck43 | $86.0 \pm 0.0$ | $0.0 \pm 0.0$ | $1.2 \pm 0.0$ | $\mathbf{98.8 \pm 0.0}$ | $84.00 \pm 0.00$ |
| Parity | $18.3 \pm 15.7$ | $\mathbf{70.0 \pm 48.3}$ | $0.0 \pm 0.0$ | $0.0 \pm 0.0$ | $11.80 \pm 11.72$ |
| Tomita 2 | $3.0 \pm 0.0$ | $0.0 \pm 0.0$ | $0.0 \pm 0.0$ | $100.0 \pm 0.0$ | $1.00 \pm 0.00$ |
| Tomita 5 | $4.0 \pm 0.0$ | $\mathbf{100.0 \pm 0.0}$ | $0.0 \pm 0.0$ | $0.0 \pm 0.0$ | $2.00 \pm 0.00$ |
| Tomita 7 | $5.0 \pm 0.0$ | $0.0 \pm 0.0$ | $\mathbf{100.0 \pm 0.0}$ | $0.0 \pm 0.0$ | $3.00 \pm 0.00$ |

**Results.** We run the $L_{nsp}^{\star}$ algorithm on 8 representative languages and track the usage of the type of disagreement used to update its hypothesis. Table 3 depicts the usage of different types of disagreements on three different sample sizes averaged across 10 runs. The prefix membership column indicates the percentage of times the prefix membership labels were used. The Cont. B1 column represents the cases where the hypothesis predicted that a certain continuation is permitted, but the continuation label indicated that such continuation is not allowed in the target DFA. The Cont. B2 column represents the cases where the hypothesis predicts that continuation is forbidden, but the true labels indicate otherwise. This is the case where the generative query oracle is invoked to generate accepting suffixes.

The results indicate that continuation labels are heavily used for all Dyck languages. Such languages also contain several transitions with dead states, and the results indicate that the algorithm is able to leverage continuation labels to identify new states. On parity and Tomita-5, the algorithm relies only on the prefix membership labels. This is natural since the DFAs for those languages do not have any dead states, and hence the continuation labels are uninformative. From positive examples, the algorithm can obtain negative information from the prefix membership labels for such languages. Both Tomita-2 and Tomita-5 have a transition to dead state. Since all these are small DFAs, the algorithm converges to the target DFA with only a few refinements.

---

**Tomita-2** $(10)^*$

You are a language generator. You will be given a regular language and you will generate one string that is in the language and end with \n (newline) character.
Language Description: The language consists of binary strings of the form $(10)^*$, i.e., repeating "10" zero or more times.

Multiple examples of strings in the language (each string ends with newline \n):
generate:10\n
generate:101010\n
generate:10\n
generate:1010\n
generate:1010101010\n
generate:101010\n
generate:

---

**Parity** (odd number of 1 symbols)

You are a language generator. You will be given a regular language and you will generate one string that is in the language and end with \n (newline) character.
Language Description: The language consists of all binary strings (alphabet $= \{0, 1\}$) with an odd number of "1" symbols. The number of zeros does not matter and the number of "1"s should be odd.

Multiple examples of strings in the language (each string ends with newline \n):
generate:1\n
generate:1101\n
generate:010\n
generate:010101\n
generate:111\n
generate:11001\n
generate:11111\n
generate:

---

Figure 8: Example prompts provided to the LLM for regular-language generation experiments. Each prompt consists of a natural language instruction describing the target regular language and multiple exemplars illustrating valid strings from that language, terminated by a \n newline character. See Sec. I for more details.

## I    EXPERIMENTS WITH LLMS

In this section, we explore the application of the $\text{L}^{\star}_{\text{nsp}}$ algorithm to extract DFAs from open-source LLMs. To mitigate the computational challenges associated with large vocabularies we restrict the vocabulary size during generation and to simplify tokenization, we use byte-level LLMs, which naturally fit in our setup. For our experiments, we adopt Meta's Byte-Level Transformer (Pagnoni et al., 2025), which operates over Unicode characters and is similar to a character-level language model.

### I.1    SETUP

**Tasks and Models.** For our experiments, we use two LLMs (1B and 7B BLT (Pagnoni et al., 2025)) and 5 regular languages. The 5 tasks include (i) *Tomita-2* $(10)^*$, which contains zero or more repeated blocks of '10', (ii) *Tomita-4*, which contains strings that do not have '000' as a substring, (iii) *Tomita-7*, which contains strings from $0^*1^*0^*1^*$, (iv) the *Parity* language, which contains an odd number of 1s, and (v) DYCK-$(1, 4)$, which contains balanced parenthesis with 1 type of bracket and depth at most 4.

Table 4: Results on extracting DFAs from 1B and 7B BLT models using $L^{\star}_{\mathrm{nsp}}$ on 5 regular languages. The NSP-Acc. column denotes the accuracy of the extracted DFA. The # States column denotes the number of states in the extracted DFA. The 'Baseline' column denotes the accuracy of predicting the NSP labels of the LLM using the DFA of the target language. The 'Erron. Acc.' column denotes the accuracy of the erroneous strings identified using the extracted DFA. See Sec. I for more details.

| Language | 1B BLT | | | | 7B BLT | | | |
| | NSP-Acc. | # States | Baseline | Erron.Acc. | NSP-Acc. | # States | Baseline | Erron.Acc. |
| --- | --- | --- | --- | --- | --- | --- | --- | --- |
| Tomita-2 | 100.0 | 10 | 0.0 | 100.0 | 100.0 | 3 | 100.0 | NA |
| Tomita-4 | 99.5 | 7 | 0.0 | 99.8 | 99.6 | 35 | 0.0 | 99.2 |
| Tomita-7 | 99.8 | 61 | 72.1 | 94.2 | 96.4 | 128 | 76.6 | 96.6 |
| Parity | 99.2 | 31 | 0.0 | 91.4 | 98.1 | 39 | 2.7 | 97.2 |
| DYCK-$(1, 4)$ | 94.0 | 53 | 22.9 | 83.2 | 97.8 | 99 | 73.3 | 85.0 |

These languages were chosen for the following reasons: Tomita-2 is one of the simplest languages and is useful to include as a sanity test; yet even a 1B model does not perfectly capture this language, as we will see later. Tomita-4 is relevant to the general use case for verification, where we would not want the model to generate something (e.g., '000', passwords, or forbidden words) and would like the algorithm to detect that. Tomita-7 checks whether the model can generate a certain alternating pattern. Parity checks modular counting behaviour and is representative of some other Tomita grammars, which also involve similar modular counting. The language DYCK-$(1, 4)$ checks bracket-matching ability with one type of bracket. As discussed later, we also tried DYCK-$(2, 4)$, but the LLMs were much poorer at generating well-balanced strings with 2 types of brackets than with 1.

**Experimental Setup.** For each task and model, we prompt the LLM to generate strings from the language. During generation, we restrict the vocabulary to the symbols of interest, which contain '(' and ')' for Dyck and '1' and '0' for the others. Apart from that, the model is asked to end each string with the newline character "\n", which serves as the [EOS] counterpart in this setup. The approach is quite simple and minimal. We provide a prompt with three parts: (i) first, a general instruction which tells the LLM that it is supposed to generate a string from the language ending with a newline, (ii) a description of the language, and (iii) in-context examples. The prompt ends with the string generate:, after which the model is supposed to generate a string from the language. A couple of example prompts are provided in Figure 8. To extract DFAs, we sample 100 NSP-labelled strings from an LLM with top-p sampling with $p = 0.9$ as a training set and evaluate the extracted DFAs on 1k freshly sampled NSP-labelled strings.

## I.2   RESULTS

**Accuracy of extracted DFAs.** The first question that we investigate is whether the DFAs extracted using the $L^{\star}_{\mathrm{nsp}}$ algorithm are accurate with respect to the NSP accuracy or error defined in Sec. 2. Note that the NSP accuracy on any example is 0 if even one of the predicted membership or continuation labels is incorrect. The NSP accuracy indicates how faithful the extracted DFA is in capturing the LLM's (truncated) support. Obtaining a DFA with good accuracy is essential before conducting further analysis with the extracted DFA.

Table 4 summarizes the overall results and reports the NSP-accuracy for each task for the 1B and 7B LLMs. As can be seen from the table, for all tasks, the NSP-accuracies of the extracted DFAs are quite high (94%–100%). Note that the accuracy of a random labelling function will be near zero, and as a baseline, we use the DFA from the target language (e.g., Tomita-2, Parity, etc.). The NSP-accuracy of the target baseline DFA denotes the accuracy that one would obtain if they used the target language DFA to predict the NSP labels of the LLMs. To avoid confusion, we emphasize that the goal of the extraction algorithm is to obtain a DFA that is faithful to the NSP labels induced by the LLM and not to the labels induced by the target DFA. As can be seen from the table, the baseline accuracy using the target language DFA is quite low for most languages, more so for the 1B LLM.

Table 5: Examples of mismatch strings found via extracted DFAs. The columns $\hat{A}_{1B}$ and $\hat{A}_{7B}$ indicate whether the string was in the support of 1B and 7B LLM, respectively. The column $A^\star$ indicates whether the string is in the target language. See Sec. I for more details.

| Language | Labels | | | Mismatch strings $x$ |
|---|---|---|---|---|
| | $\hat{A}_{1B}$ | $\hat{A}_{7B}$ | $A^\star$ | |
| | 1 | 0 | 0 | 1 0 1 0 1 0 |
| Tomita-7 | 1 | 0 | 0 | 1 0 1 0 1 1 0 |
| | 1 | 0 | 1 | 0 0 0 0 0 0 0 |
| | 1 | 0 | 0 | 1 1 1 1 1 1 0 |
| Parity | 1 | 0 | 0 | 0 0 1 1 1 1 1 0 0 |
| | 1 | 0 | 1 | 0 0 1 1 1 1 1 0 0 1 |
| | 1 | 0 | 0 | ( ( ( ) ) ) |
| Dyck-$(1, 4)$ | 1 | 0 | 0 | ( ( ( ) ) ) ( ) |
| | 1 | 0 | 0 | ( ( ( ( ( ) ) ) ) ( ) |

We also explored DYCK-$(2, 4)$, but the generation accuracies of the LLMs were quite poor compared to DYCK-$(1, 4)$. While over 90% of the strings that the 7B LLM generates are in DYCK-$(1, 4)$, for DYCK-$(2, 4)$ this fraction was less than 30%. We ran the $L^\star_{nsp}$ algorithm for DYCK-$(2, 4)$, and the number of states blew up: the algorithm was not able to find a consistent DFA even after extracting over 1k states. We tested an extracted DFA with about 384 states, which was partially consistent with the training set, and its NSP-accuracy on the test set was about 64%. This scenario falls into the regime discussed in Sec. 7 and Sec. H.3, where the model is far from perfect and the number of underlying states is too large for our current implementation to handle.

**Identifying Erroneous Examples.** We also evaluate the effectiveness of the extracted DFAs in identifying erroneous examples. For each task, we use the extracted DFA $\hat{A}$ to find strings in the symmetric difference between the hypothesis $\hat{A}$ and the target language $A^\star$, as described in Sec. H.2. We find the first 500 strings in length-lexicographic order and report the correctness of the identified examples, which indicates the percentage of times that a string identified using the product DFA is actually erroneous.

Table 4 depicts the accuracy of the erroneous examples identified by the extracted DFA. For Parity and the Tomita languages, the erroneous strings identified by the extracted DFA are quite accurate. For DYCK-$(1, 4)$, the accuracies are still quite high but seemingly lower than for the other languages. Upon further analysis, we believe that the difference is due to the following reason: most of the strings in the training set for $L^\star_{nsp}$ are quite short (length $< 20$). When we identify erroneous strings in length-lexicographic order, the number of erroneous strings of length less than 20 is quite low (unlike Parity and the other languages), and the DFA identifies many strings that are of greater length than those seen during training. The accuracy of these strings is not as high, since they could be considered out-of-distribution as well. If we look at the first 100 erroneous strings identified by the product DFA, then over 90 of them are correctly predicted as erroneous by the extracted DFA. Additionally, we emphasize that finding strings from the product DFA is quite cheap, and even with an accuracy of $50\%$, it is much faster to generate erroneous strings from the product DFA and verify them using the LLM, compared to sequentially generating many strings from an LLM and checking whether they are erroneous.

Among the languages we tested, Tomita-2 is the only language where the 7B LLM seems to perfectly capture the support. The support of the 1B LLM is also quite close to the target, since the symmetric difference between the extracted DFA and the target contained only two strings. In all other languages in Table 4, the NSP-accuracies of the target DFA are relatively far from the support, and various erroneous examples could be found. We also tested the sensitivity to the top-p parameter $p \in \{0.8, 0.9, 0.95\}$. We found that the 1B LLM was relatively more sensitive to this parameter than the 7B LLM. For instance, with $p = 0.95$, the DFA extracted from the 1B BLT model for Tomita-2 had over 500 erroneous strings, as opposed to just 2 for $p = 0.9$.

**Model Comparison.** We investigate whether DFAs extracted from the 1B and 7B models can be used to identify differences in the behaviour of the two LLMs. In this analysis, we consider Tomita-

7, Parity, and DYCK-$(1, 4)$. For each language, we proceed as follows. We take the two extracted DFAs, $\hat{A}_{1B}$ and $\hat{A}_{7B}$, and construct the product DFA $\hat{A}_{1B} \times \hat{A}_{7B}$, which represents the symmetric difference language as described in Sec. H.2. We then generate 50 strings in length-lexicographic order from the product DFA for which $\hat{A}_{1B}(x) \neq \hat{A}_{7B}(x)$. We inspect these mismatch strings to find systematic patterns and then check whether any identified pattern holds more generally using the extracted DFAs. While this approach is not the most efficient, we adopt it because the extracted DFAs, as well as the product DFAs, have many states and are largely uninterpretable visually.

**Observations.** Table 5 shows representative examples from the symmetric difference between the 1B and 7B LLMs for each language. Based on the extracted DFAs and the mismatch strings, we make the following observations. The 1B model is generally overly permissive, whereas the 7B model is overly conservative, in the sense that the 1B model includes strings in its support that are not in the target language. Unsurprisingly, the 7B model is more accurate than the 1B model, but the cases where the 7B model is incorrect are those where it excludes strings that should be in the support. Beyond this, we find systematic differences for Tomita-7 and DYCK-$(1, 4)$, but are unable to identify any clear pattern for Parity.

*Tomita-7.* For Tomita-7 ($0^*1^*0^*1^*$), the language requires the string to alternate between 0 and 1, starting with 0, with at most 3 alternations. The 1B model contains many strings with more alternations (see the first two examples in Table 5). The 1B model also has strings of the form $1^*0^*1^*0^*$ in its support. Upon seeing such mismatch strings, we checked the extracted DFA $\hat{A}_{1B}$ and observed that such strings lead to an accepting state in $\hat{A}_{1B}$. The 7B model correctly excludes such strings from its support. Out of the 50 strings, the 1B model is incorrect in $\approx 60\%$ of the cases. The strings where the 7B model is incorrect are typically those that have 0 or fewer than 3 alternations, e.g., 00000, which belongs to the language $0^*1^*0^*1^*$ but is excluded by the 7B model.

*Dyck-(1,4).* For DYCK-$(1, 4)$, the 7B model is substantially more accurate than the 1B model. All first 50 mismatch strings correspond to cases where the 1B model makes a mistake. Looking at the next 50 mismatch strings, the 1B model is incorrect on 49/50 of them. We find two patterns in the mistakes made by the 1B model. First, it includes strings in its support with unclosed brackets, i.e., prefixes of DYCK-$(1, 4)$ that are missing one or two closing brackets (see the first two examples in the DYCK-$(1, 4)$ row in Table 5). The model does not seem to make mistakes of the form ( ) ) where the number of closing brackets exceeds the number of opening brackets. Second, the 1B LLM sometimes fails to enforce the depth constraint and includes strings with depth $> 4$ in its support (see the last example in Table 5) while the 7B LLM correctly excludes them.

