# OpenReview forum: "Automata Learning and Identification of the Support of Language Models"
_ICLR.cc/2026/Conference — ICLR 2026 Poster_

### Official Review · Reviewer_4bT5 · 2025-10-21

**Soundness:** 4
**Presentation:** 4
**Contribution:** 4
**Rating:** 8
**Confidence:** 4

**Summary:**

The paper proposes a next-symbol-prediction-based formal language learning framework with applications in extracting formal descriptions of neural language models. This framework augments Gold’s seminal language learning framework with information about the allowed string continuations. This allows much larger classes of languages to be learnable from positive examples alone. Nevertheless, the authors show that the added information still does not enable efficient PAC learning of finite-state automata (FSAs), but in some cases allows for more efficient L*-style learning based on membership and equivalence queries.
The L* learning framework is compatible with using a language model as an oracle, which enables the authors to use the introduced algorithms to extract FSAs from trained language models. They find the algorithm to be very sample-efficient. Moreover, the extracted FSAs often agree with the ground-truth (whenever the neural model is well-trained), unless the language is hard to learn (like parity), as expected.

**Strengths:**

- The paper is very well structured and clearly written. The contributions are clear and in my opinion important.
	- Most significantly, the paper takes a well-studied and known learning setting and updates it to a well-motivated and relevant setting useful in modern language models
- The results are theoretically thoroughly backed and explained with well-composed proofs.
- Although the main contributions are theoretical, the paper provides extensive and well-justified empirical validation of the results with languages well-studied in the theoretical literature on neural networks and formal languages.
	- These experiments show the practical improvements of the proposed algorithm, which is also theoretically shown to be much more efficient in some cases
- I found the FAQ section to be useful.
- It is useful that the algorithm is completely specified and described understandably in pseudocode.

**Weaknesses:**

- The work is very well written and presented. I found no major weaknesses.
- I just had one thought: I imagine looking at automata learned from some actual large-ish-sized language models would be interesting—maybe specifically domain-specific or domain-prompted large LMs. I imagine this is not possible with the current algorithmic complexities?

**Questions:**

- How does one (gracefully) handle learning states that are technically learned by the language model but not in the truncated support?
- How does one choose the truncation threshold for the support of a language model?
- Any intuition of what happens when the language is not actually regular? The number of states just keeps increasing?
- Are there any convenient descriptions of languages where NSP information provably makes learning more efficient (even though it, in general, does not hold)?
- I believe that, technically, it can hold that a symbol can be in the set of permissible continuations in a neural LM but never lead to a string in the support if the generation never halts (so-called improper or non-tight language models). Is that intuition correct? If so, should this condition be enforced? I suppose this is equivalent to enforcing a finite expected length, which you do?
- Two nit-picks:
	- Is the framing of “positive examples only” completely justified? As mentioned, each (positive) example also brings with it possibly many negative examples (prefixes). This is purely a naming thing.
	- If the language model is regular (as assumed), the distribution over negative examples should also be regular (and thus defined and available)

---

> ### Author Response · Authors · 2025-11-21
>
> Thank you for your thoughtful comments and time.
>
>
> > I imagine looking at automata learned from some actual large-ish-sized language models would be interesting—maybe specifically domain-specific or domain-prompted large LMs. I imagine this is not possible with the current algorithmic complexities?
>
> Yes, as discussed in the paper, the algorithm currently cannot be applied to actual LLMs in a plug-and-play fashion primarily due to the computational challenges associated with vocab size (leading to many more MQ calls) and tokenization schemes. However, we are currently trying out some ideas to test LLMs prompted to generate regular languages. Given the computational challenges and the need to try out experimental design choices, it is a bit difficult to obtain results in a couple of weeks, but we are hoping to have some preliminary results before the end of the discussion period.
>
>
>
> ### Questions
>
>
> > How does one (gracefully) handle learning states that are technically learned by the language model but not in the truncated support?
>
> The question is unclear to us, and it would be helpful if you could elaborate on this. If you are talking about states when the sampling is done without any truncation, then it is trivially the single state DFA with self-loops that accepts all strings $\Sigma^*$ since softmax assigns a nonzero probability to all tokens in the vocab.
>
>
>
> > How does one choose the truncation threshold for the support of a language model?
>
> In practice, if one is planning to serve their language model with top-p with p=0.9, then they should choose the truncation threshold accordingly; based on the threshold they will use to generate from the LM.
>
> For our experiments in a controlled setting, the situation is different, and we did the following: to train Transformers for any regular language, we are currently generating strings by constructing a PDFA as described in Sec. 6 (Lines 388-391). In simple terms, you could say that at every step, the probability of the next token is divided uniformly among the permissible tokens. If the end-of-sequence token is allowed, then it is sampled with some probability $t$, or else the same strategy among other tokens is followed to sample the next token/character.
>
> We know that for the target PDFA, any threshold > 0 and < 0.1 should work for all the languages considered. For Transformers, if we choose an extremely small threshold, then it allows all symbols (because of softmax). We tried {0.05, 0.01, 0.005}. We didn’t find much difference in preliminary experiments and stuck to 0.05 for all our experiments.
>
>
> > Any intuition of what happens when the language is not actually regular? The number of states just keeps increasing?
>
> Yes, DFAs can capture languages that can be recognized with a finite amount of memory. Any non-regular language has some form of unbounded memory (e.g., Context-free languages have an unbounded stack, counter languages have unbounded counters, etc). For instance, bounded Dyck languages are regular since the depth is bounded. The number of states grows exponentially with the depth. They cannot be recognized with DFAs with a fixed number of states. If one assumes that they will only get strings of bounded length in practice, then the underlying language will be regular, although the number of states can be very large.

---

> > ### Author Response · Authors · 2025-11-21
> >
> > > Are there any convenient descriptions of languages where NSP information provably makes learning more efficient (even though it, in general, does not hold)?
> >
> > We believe this is a very interesting question, one for which, unfortunately, we do not yet have an interesting concrete answer. As you mentioned, it does not hold in general. We hypothesize that it makes learning more efficient when there are more transitions to the dead state, which is not unnatural since languages (programming language, natural language, etc) have a lot of structure and often certain types of symbols are not allowed. One direction we explored was to see if we could prove better sample complexities based on the number of paths from the start state to the accepting states. For instance, the class of Conjunctions has only one path and hence can be learned efficiently (L219-226), whereas its counterpart, Disjunctions, has many paths and hence does not lead to any improvement with NSP labels. But we haven’t been able to show anything formally for this problem, beyond the simple cases of conjunctions and disjunctions.
> >
> >
> > > I believe that, technically, it can hold that a symbol can be in the set of permissible continuations in a neural LM but never lead to a string in the support if the generation never halts (so-called improper or non-tight language models). Is that intuition correct? If so, should this condition be enforced? I suppose this is equivalent to enforcing a finite expected length, which you do?
> >
> > Yes, the fact that the model shouldn’t generate strings of infinite length is a reasonable assumption in our opinion. This case is discussed in App H.3. We didn’t encounter this issue in any of our experiments, and we believe that since language models are trained to predict [EOS] after a certain number of steps, they eventually do so. It follows from the assumption that the support is a regular language since regular languages can have strings of finite length (even if the set itself can have an infinite number of strings).
> >
> >
> >
> > >Two nit-picks:
> > > Is the framing of “positive examples only” completely justified? As mentioned, each (positive) example also brings with it possibly many negative examples (prefixes). This is purely a naming thing.
> >
> > We believe it is largely justified. In our setting, the focus is to learn from *strings in the language* (positive examples). Information about negative examples is provided with additional labels, but care should be taken to ensure that they are sufficient. For instance, without the continuation labels, as described in Sec. 3 (L194-201), identifiability is not possible, and the equivalence query is not well-defined, which is necessary for exact learning to be feasible.
> >
> > As mentioned in Sec.1 (L77-79), the focus is on generative models where the strings we generate are positive/in the language (of the generative model) and there is no natural distribution over negative examples (more on that in your next question below). Since just positive examples are insufficient, the formulation of the proposed learning model is designed based on the additional information we can obtain from a language model, which turns out to provide sufficient information about the negative examples in the sense of identifiability. We never claim that there is no information about negative examples, which is, of course, not feasible. Rather, the goal is to learn from positive examples along with additional labels (which could have information about negative examples) such that the guarantee that is obtained is also with respect to a desired distribution. We agree that it can be slightly confusing. We will explicitly state that information about negative examples can be obtained from membership and continuation labels.
> >
> >
> > > If the language model is regular (as assumed), the distribution over negative examples should also be regular (and thus defined and available)
> >
> > If the support of the language model is regular, then the complement of its support would also be regular (strings not in support). But we don’t think we have a natural distribution over those strings that is available to us. For the strings in the language, we can just sample from the language model and get a guarantee with respect to that distribution over strings. But it is unclear how we can sample negative strings in a natural way. For instance, consider PDFAs that define a distribution over strings and have a regular support (induced by the corresponding DFA). While we can easily get the complement of the support language and its corresponding DFA, there is no natural way to get a PDFA for its complement. So the distribution over negative examples will be inherently artificial.

---

> > > ### Comment · Reviewer_4bT5 · 2025-11-24
> > >
> > > Thank you very much for the thorough explanations. They answer my questions very well!
> > >
> > > Regarding my unclear question: What I (*believe*; I apologize, it has been a while, and I admit the question is so vague even I can't fully decipher it anymore) wanted to ask is how you would imagine handling the scenario where, e.g., a threshold of 0.1 would be used in experiments, but maybe a state would only be discovered/added to the set of states with a lower threshold of 0.01. I think you answered this in one of the other answers, where you tried multiple thresholds.
> > >
> > > I maintain my positive evaluation of the work, and I hope it is accepted.

---

### Official Review · Reviewer_xFFN · 2025-10-28

**Soundness:** 3
**Presentation:** 3
**Contribution:** 3
**Rating:** 6
**Confidence:** 4

**Summary:**

The paper formalizes support identification for language models in an NSP supervision setting that mirrors LM decoding, establishes identifiability of minimal DFAs from positive NSP data, proves worst‑case hardness results that survive NSP labels, and proposes $L^\*_{nsp}$, an extension of Angluin’s $L^\*$ that uses LM‑simulable queries (membership and prefix‑conditional generation). Experiments with Transformer teachers on 11 regular languages demonstrate that modest positive NSP data suffice to recover the target or teacher‑support DFAs and to detect erroneous strings when teachers deviate.

**Strengths:**

1. NSP exactly captures what a practitioner can extract from an LM under top‑k/top‑p/min‑p decoding. The focus on support (rather than probabilities) is natural and practically important.
2. $L^\*_{nsp}$’s Lemma 5.1 is a neat reduction from continuation mismatches to standard counterexamples using at most one generative query. This method does not require a perfect equivalence oracle which is often infeasible in practical.
3. The evaluations are careful and complete. The method recovers large Dyck DFAs with tens–hundreds of positives and surfaces nontrivial errors. The ablations support the claim that continuation labels are particularly beneficial on languages with dead‑state transitions.

**Weaknesses:**

1. The regular-language focus is reasonable for support identification, but large vocabularies are the key obstacle for LM-scale studies. While the paper is honest about this, many NLP applications involve **huge vocabularies**. The closure step’s $(O(|Q||T||\Sigma|))$ cost is problematic for realistic LMs, and the paper does not yet provide system‑level accelerations. If the authors can solve this concern I am glad to raise my score.
2. The PAC guarantee is **with respect to the teacher LM’s own distribution**. This is appropriate for support identification but complicates cross‑teacher comparisons and may hide rare but catastrophic support errors (if the teacher rarely visits those prefixes). (Section 5.)
3. $L^\*_{nsp}$ sometimes needs one **generative** query to produce an accepting suffix. If the teacher’s support is not regular or exhibits long nonterminating paths under a truncation rule, the procedure may stall (H.3).

**Questions:**

1. How sensitive is the learned support DFA to the choice and hyperparameters of truncation (top‑k vs. top‑p vs. min‑p)? Could the same teacher yield meaningfully different supports across rules?
2. If the true support is not regular but “close,” does $L^\*_{nsp}$ converge to a minimal DFA that best approximates NSP labels in expectation (e.g., a smallest consistent DFA on observed NSP data)?

---

> ### Author Response · Authors · 2025-11-21
>
> Thank you for your thoughtful comments and time.
>
>
>
> > (W1) The regular-language focus is reasonable for support identification, but large vocabularies are the key obstacle for LM-scale studies.
>
> As you mentioned, we have been very clear about this in the paper (Sec. 7, FAQ Q1, App H.3). The goal of this work was to introduce a new model of learning tailored to LMs and characterize learnability theoretically. The positive result is that polynomial-time learnability with generative queries is possible, although it incurs $O(|\Sigma||Q||T|)$ complexity inherited from L*. As a first work in this direction, we think it is reasonable and is a precursor to more efficient algorithms.
>
> There are two other points to note.
>
> (i) While the closure step is computationally demanding, we think the efficiency concerns can be mitigated to a large degree in practice. Firstly, we did implement some efficiency improvements compared to a naive version. Many of the membership query (MQ) calls in the closure step are repetitive, and to avoid redundant computation, we have an LRU cache that stores the last $10^5$ queried strings and avoids redundant computation. Secondly, for applicability to large-scale LMs with a larger vocabulary, another immediate direction is to parallelize the computation in the closure step. The checks in the closure step need not be in any particular order and hence can be processed in batches, which could lead to significant improvements if one has GPUs with decent memory. This wasn’t of immediate concern to us since our goal was to test the $\mathrm{L^{\star}_{NSP}}$ algorithm in a controlled setting, and our GPUs cannot maximize them anyway. We will discuss these points in the next version of our draft in more detail, and we will also include the code during the discussion period and clearly specify where the LRU cache is.
>
> Given the focus and objective of our work, we believe that significant engineering improvements beyond these will be outside the scope of this work.
>
>
> (ii) The $|\Sigma|$ factor is likely to be an inherent bottleneck while learning DFAs. The L* algorithm was published about 40 years ago, and several improvements and analyses building on it have been published. While the other constants have been improved, it seems that $|\Sigma|$ factor in queries is unavoidable [1, 2] based on currently known lower bounds.
>
> [1] Lower Bounds for Active Automata Learning. L. Kruger et al. ICGI 2023.
>
> [2] Algorithms for learning finite automata from queries: A unified view. J.L. Balcazar et al. Advances in Algorithms, Languages, and Complexity. 1997
>
>
>
> > (W2) The PAC guarantee is with respect to the teacher LM’s own distribution. This is appropriate for support identification but complicates cross‑teacher comparisons and may hide rare but catastrophic support errors (if the teacher rarely visits those prefixes).
>
> We believe that the guarantee with respect to the teacher LM’s own distribution is a positive aspect. Typically, one can simulate the L*s EQ+MQ algorithm by simulating the equivalence query by sampling strings from a distribution. However, the desired distribution is rarely available in practice. In this scenario for verification, the language model’s own distribution is the most likely desired distribution. Additionally, while it is framed that way, the guarantee is not strictly restricted to just the LM’s own distribution. If the algorithm receives a set of NSP labelled examples from *any* distribution which has the *same* support as the support of the LM’s distribution, then the algorithm will work as is. The guarantee is, of course, probabilistic, so worst-case scenarios can generally not be addressed with PAC bounds. Also, cross-teacher comparisons, while potentially interesting, are not the focus of this work.
>
>
>
> > (W3) L*-NSP sometimes needs one generative query to produce an accepting suffix. If the teacher’s support is not regular or exhibits long nonterminating paths under a truncation rule, the procedure may stall (H.3).
>
> As you have mentioned, this is clearly stated in the paper and also follows from the problem formulation and the assumptions that the target language is regular. We believe that in practice, we would not be dealing with infinite-length strings, so this assumption is reasonable. Additionally, as mentioned in App H.3, we never found this to be an issue in our experiments. We believe that since the models in practice are trained to produce [EOS] after a certain number of steps, they naturally tend to produce [EOS] tokens after a certain point (unless they are explicitly trained not to produce them, which would violate the assumption).

---

> > ### Author Response · Authors · 2025-11-21
> >
> > ### Questions
> >
> > > How sensitive is the learned support DFA to the choice and hyperparameters of truncation (top‑k vs. top‑p vs. min‑p)? Could the same teacher yield meaningfully different supports across rules?
> >
> > They are, of course, sensitive to some degree, but within reason based on our observations. Top-p and min-p are very similar conceptually; we only use min-p since it intuitively fits directly here. If k in top-k is the same as $|\Sigma|$ or the threshold in min-p/top-p is too lenient, then the support will include all strings (1-state DFA) since softmax assigns a nonzero probability to all tokens and will be uninformative. If k=1 or the threshold in min-p and top-p is too strict, then the generation is equivalent to greedy decoding, and the support will have only 1 string. In our experiments, we explored min-p with p = {0.05, 0.01, 0.005}; we didn’t find any notable difference between them, and then stuck to 0.05.
> >
> >
> > > If the true support is not regular but “close,” does L*-NSP converge to a minimal DFA that best approximates NSP labels in expectation (e.g., a smallest consistent DFA on observed NSP data)?
> >
> > It is unclear how ‘close’ can be clearly defined here. Any support that is not regular must use some form of unbounded memory (unbounded stack, counters, etc). Analyzing learnability with such computational models then becomes difficult since it may not be computationally feasible unless we add strong assumptions.
> >
> > There are some consequences, though. For instance, if the true support is Dyck-2 (context-free) instead of Dyck-(2, k) (bounded depth k), then in practice, we will still receive strings of finite length L which will have depth at most L/2. The algorithm in that case can learn the DFA corresponding to Dyck-(2, L/2), and since it will be a consistent learner, it can achieve low NSP error w.h.p. on unseen examples depending on the number of training samples. More generally, if we are working with practical LMs that have a fixed context length or a maximum generation length, then the support will have strings of bounded length and the support language will always be regular, though the corresponding DFA could potentially have a very large number of states.

---

> > > ### Author Response · Authors · 2025-11-25
> > >
> > > We have updated the paper with a discussion on avenues for efficiency improvements in Sec. H.3. We have also uploaded our code along with a pointer to the LRU cache for your reference. Please see the general response or the paper for more details.

---

### Official Review · Reviewer_VJGS · 2025-10-29

**Soundness:** 4
**Presentation:** 4
**Contribution:** 3
**Rating:** 6
**Confidence:** 5

**Summary:**

This paper introduces a new model for automata learning where the queries the learner can ask are the ones that are more naturally provided by a language model that exposes token logits at each step. That is, in normal automata learning, one can only ask membership queries (is this string in the language) and equivalence queries (did I find the right automaton, if not give me a counterex). Whereas the authors propose a setting where one can, given a sequence a1...an ask for every token b, whether a1..aib is in the language. This is a very powerful oracle that can provide many positive and negative examples at once. The paper shows that even with this powerful new model identification in the limit remains hard. They therefore move to a setting where the learner can ask the LM itself to generate positive sequences (together with their possible continuations). This approach yields a PAC guarantee (as long as equivalence is checked on the same distribution of the LM). The empirical analysis shows small DFAs can be relearned from a transformer (that were trained to recognize them).

**Strengths:**

- The formalization of NSP queries is a clean one.
- The paper studies a question worth studying and though the hardness result is somewhat unsurprising (and also the generative one since very similar results hold for learning DFAs when strings are sampled from a distribution to mimic an equivalence oracle), it is nonetheless an interesting result that NSP queries help learning faster empirically as shown (i.e., if one has the examples handy, might as well use them).
- The paper is formal and self-contained

**Weaknesses:**

- My main criticism of the paper is that open source LLMs provide the perfect setting for evaluating interesting questions and this paper instead has gone for training small transformers on toy languages. Before reading the eval description in the introduction, I was expecting this paper to do the following
1. Prompt the LLM (any open one) with the query "Generate strings that are accepted by the regular expression R"
2. Use L*NSP to learn what the LLM generate
3. Compare automaton to R
Then the evaluation would tell us whether 2B parameter models learn worse than 8B parameter models, for example, and what type of mistakes they make. I would love to see such an evaluation. One really interesting aspect would be to see when using (Real-world regexes) what characters LM "prefer" within character classes and how they fail to cover them
- Related to the above problem is that the paper does not seem to discuss the fact that LLMs produce tokens and not characters.

**Questions:**

Can the authors do the experiment I discuss in weakness for some interesting regular expressions and small open LLM? Ideally with some practical regexes than Dick languages (maybe pick 10 from regexlib or automatark)

How does the algorithm change when moving from characters to tokens?

Why are there EOS tokesn between characters in 393? IS this a way to force a certain tokenization scheme?

MINOR COMMENTS:
- line 145: missing _L in phi
- what does well-defined mean on l 210?
- Spell out PDFA in line 388 and point to a reference
- is the reference to Sec 2.2 in line 400 incorrect?
- maybe relevant paper to cite https://arxiv.org/abs/2501.02825

---

> ### Author Response · Authors · 2025-11-21
>
> Thank you for your thoughtful comments and time.
>
>  **Regarding experiments with LLMs** *(Weakness 1 and Question 1)*
>
> **Why small Transformers?** First, we would like to clarify why we focused on experiments with Transformers trained from scratch. The focus of our experiments was to test the efficacy of the algorithm itself, which naturally precedes applying the algorithm to understand other objects like LLMs. To evaluate whether the algorithm consistently produces low-error hypotheses, to understand its sample efficiency, and whether it uses the additional labels in the NSP setting, we had to evaluate it in a more controlled setting where we have perfect or near-perfect oracles and the ground-truth DFA. We believe conducting such evaluations and ablations for the algorithm is an essential step before applying it to understand other models.
>
> **LLMs.** While we do not see any fundamental barriers to adapting/applying the L*NSP algorithm to LLMs, there are some concrete challenges that, at present, prevent a direct “plug-and-play’’ application. Most of these are already discussed in the paper (Sec. 7 Limitations, FAQ Q1, App. H.3); we expand on them here.
>
> *Query complexity and vocab size.* The algorithm is efficient in a polynomial-time sense, but as mentioned in Sec. 7 and H.3, in practice, a key bottleneck of L* and L*NSP is the closure step, which uses ($\Omega(|T||\Sigma|)$) queries every time a new state is added. Even for a single counterexample, this step can be invoked many times, depending on the number of states in the target DFA. In our experiments with Transformers trained from scratch, the factor |\Sigma| is small and therefore not dominant. For an actual LLM, however, |\Sigma| corresponds to a vocabulary of tens of thousands of tokens, so the closure step would require a much larger number of queries.
>
> *Cost per query.* Moreover, the |\Sigma| factor directly counts the number of membership queries, i.e., the number of times the language model is called (each call being a forward pass). The Transformers we train from scratch are relatively small, so these calls are relatively cheap. In contrast, even for a 1B-parameter LLM, each call would be more expensive, in addition to the orders-of-magnitude increase in the number of queries due to the larger vocabulary.
>
> *Imperfect Teacher.* As mentioned in Sec. 7 and App. H.3, when the teacher LM is far from perfect on the target language, the number of underlying states can be huge. In such cases, the algorithm initially discovers many states rapidly and then slows down significantly when (|Q| > 500), because each closure step requires a large number of queries. Since the LLMs are not primarily trained to generate from a regular language, they are likely to be imperfect, and this scenario could plausibly occur.
>
>
> We believe these challenges can be mitigated to a large degree with additional work (primarily in parallelizing the closure step), and we already have certain caching mechanisms in place that improve efficiency over a naive implementation. However, the bottleneck is significant enough to prevent one from simply prompting an LLM and directly running L*NSP to completion.
>
> *Ongoing experiments.* That said, we are currently setting up experiments and actively evaluating several design choices to test the algorithm with LLMs, in line with the reviewer’s suggestions. We do agree with the reviewer that it is quite relevant, but given the need to try different experimental design choices and the computational cost associated, it may not be feasible to obtain solid results during the discussion period. Nonetheless, we are hoping to have some preliminary results before the end of the discussion period.
>
>
> Finally, we would like to emphasize that our focus was on introducing a novel and relevant model of learning, providing a theoretical characterization of learnability (identifiability, hardness, and learning with generative queries), as well as showing that it is not only an abstract model, but it is an effective language model-based oracle. We acknowledge that certain computational challenges are a limiting factor at the moment (clearly acknowledged in the paper), and we strongly believe that making progress in addressing the computational challenges associated with the algorithm can allow us to explore various interesting questions about the behaviour of LLMs.
>
>
>
>
> **Character to tokens** *(Weakness 2 and Question 2)*
>
> The algorithm is not specific to characters but to an alphabet or vocabulary of distinct symbols. In that sense, moving from characters to tokens does not change the algorithm: we simply treat tokens as the alphabet $\Sigma$. The main effect is practical, not conceptual: the vocabulary size for LLMs is much larger, so all bounds and costs that scale with $|\Sigma|$ (e.g., in the closure step) become much larger, as discussed above and in the paper (Sec. 7, H.3).

---

> > ### Author Response · Authors · 2025-11-21
> >
> > ### Questions
> >
> > > Why are there EOS tokens between characters in 393? IS this a way to force a certain tokenization scheme?
> >
> > The [EOS] tokens are between strings and not characters. While we only care about the generation of the first string, we provide multiple strings in one example sequence during training so that a batch of examples has sequences of similar lengths. As described in the paper, the strings are generated using a PDFA, and we have a geometric distribution over lengths. Thus, many of the strings can have small lengths, and to ensure that the model sees sequences of lengths around 256 (context length), we generate multiple strings and concatenate them together, separated by [EOS]. This is similar to how language models are trained in practice and provides additional signal during training.
> >
> >
> > > line 145: missing _L in phi
> >
> > Thanks!
> >
> > > what does well-defined mean on l 210?
> >
> >
> > For exact learning with equivalence queries (EQ) or EQ+MQ to be feasible, it is necessary for the EQ oracle to be well-defined in the following sense: With respect to any target DFA $A$, the oracle $EQ(\cdot; A)$ should be able to take any hypothesis $\hat{A}$ and return a counterexample $x$ such that the labels of $A$ and $\hat{A}$  disagree on $x$, or it should return a True (or match) when they are isomorphic and represent the same language. As discussed in Lines 194-201, if we only have positive examples or just the prefix membership labels, then such an equivalence query oracle does not exist or is not well-defined. This is because, in that scenario, two DFAs $A$ and $\hat{A}$ could be distinct, but the EQ with respect to $A$ cannot always produce a counterexample to differentiate them. Thus, with just positive examples, both membership and continuation labels are necessary for the equivalence query oracle to be well-defined and for identifiability.
> >
> >
> > > Spell out PDFA in line 388 and point to a reference
> >
> > Thanks, we will spell it out, and they are described in more detail in App. C (preliminaries).
> >
> > > is the reference to Sec 2.2 in line 400 incorrect?
> >
> > It is correct. Sec 2.2 describes formally how a language model with top-p/min-p sampling can act as an oracle and relates to NSP.
> >
> > > maybe relevant paper to cite https://arxiv.org/abs/2501.02825
> >
> > Thanks! Looks relevant, we will add a note in the next version.

---

> > ### Comment · Reviewer_VJGS · 2025-11-22
> >
> > Thanks for the clarifications. I remain in favor of acceptance.
> >
> > Some suggestions for the implementation problems mentioned by the authors.
> > I recommend looking at constrained decoding libraries (e.g., llguidance, and GreatGramma, Park et al ICML25) to see how these implementations process all tokens efficiently at once against a constraint to see if a completion is possible. These approaches will give a way to speed up the oracle  and closure algorithms and avoid iterating over all symbols.
> >
> > I would like to push back against the comment stating that "we simply treat tokens as the alphabet $\Sigma$". This is in general not a sound way to approach the problem in my opinion. If the string "aa" could be tokenized as either "a""a" or "aa" and I'm interested in whether the LM has learned the language (aa)*, the approach will need to take into account the different tokenization schemes and the fact that perhaps one tokenization has low probability and one has high probability. If I understand correctly, the min-sampling approach in this case would infer that a sequence could be both a negative and positive example and one would need to resolve this discrepancy somehow. Am I misunderstanding something?

---

> ### Author Response · Authors · 2025-11-24
>
> Thank you for your response and for the suggestions; they seem relevant!
>
> Regarding your point about tokenization, we think you have raised a valid point and we would like to elaborate and provide further clarification. For word-level and character-level LMs, we believe we are on the same page. Regarding subword or BPE tokenizers, we agree that the semantics are different, and we discuss it in more detail below.
>
> In our formal setup, the alphabet $\Sigma$ is arbitrary, and all strings are sequences over $\Sigma$. When we apply the framework to an LM, we take $\Sigma$ to be the LM’s *token* vocabulary, so `["a","a"]` and `["aa"]` are two different strings in $\Sigma^{\ast}$. The NSP labels, membership queries, and the truncated support in the paper are all defined on such token sequences. From this token-level perspective, there is no single sequence that is both a positive and a negative example: `["a","a"]` and `["aa"]` may receive different labels, but they are different elements of $\Sigma^{\ast}$. While `a` and `aa` overlap in surface form when we read them as characters, they correspond to distinct token IDs and are treated as distinct symbols by the LM.
>
> For your concrete example, suppose the tokenizer’s vocabulary is $\Sigma = \{\texttt{a}, \texttt{aa}\}$ and the training data for the character-level language $(aa)^*$ is first converted to tokens by this tokenizer, so that all training strings could become sequences like `["aa","aa", ...]`. The LM may then assign high probability to `aa` and very low probability to `a`. Under a per-step truncation rule (e.g., min-p truncation), the token `a` might never appear in the admissible-next-token sets. In that case, the token-level truncated support contains sequences such as `["aa","aa",...]` but not sequences containing `a`, and the extracted DFA will have transitions on `aa` but effectively send `a` to a dead state. We think this is faithful to how the LM models the symbols and arguably desirable: the fact that a single `a` is essentially disallowed is a property of the trained LM under this tokenizer, and it is better for the extracted DFA to show this behavior explicitly.
>
> The resulting DFA is therefore a DFA for a regular language over the token alphabet $\{ \texttt{a}, \texttt{aa} \}$. If one thinks in terms of a character-level target language such as $(aa)^*$ over the alphabet $\{a\}$, there is no longer a one-to-one correspondence between token sequences and character strings: both `["aa"]` and `["a","a"]` can correspond to the same character string `"aa"`, and the LM may treat these two token sequences very differently. The algorithm is designed to recover the LM’s own token-level truncated support under a fixed tokenizer and truncation rule. On the flip side, we agree that this token-level framework is not directly compatible with the character-level regular expression setting you describe: the learned DFA is defined over the model’s token vocabulary and therefore characterizes its behavior over token sequences, not over character strings, so multiple tokenizations of the same string can lead to different behavior. Our earlier comment about the algorithm remaining unchanged was meant only in this token-level sense, not that the framework directly captures character-level regex behavior under subword tokenization.
>
>
> We do agree that this distinction between token-level and character-level semantics is important for subword tokenizers, and merits a separate discussion/clarification in the paper. We will include this point in the next version. For our experiments with LLMs, we are using 1B and 7B byte-level LLMs, which are similar to character-level LLMs and fit more directly. We are planning to update the paper with some results very soon.

---

> > ### Comment · Reviewer_VJGS · 2025-11-24
> >
> > Thanks. We are on the same page. I will update my score

---

> > > ### Author Response · Authors · 2025-11-25
> > >
> > > We have added some experiments with LLMs in the revised version and also included a discussion regarding the tokenizers. Please see the general response or the paper for more details.
> > >
> > > We thank the reviewer for their constructive comments and engagement, which helped improve the paper.

---

### Official Review · Reviewer_3o3h · 2025-10-30

**Soundness:** 4
**Presentation:** 3
**Contribution:** 4
**Rating:** 8
**Confidence:** 3

**Summary:**

The paper is a theoretical and empirical contribution to understanding language models learnability of formal languages by means of extracting automata from them. The data is regular languages with NSP labels (given a prefix, its acceptance as a prefix and the possible next symbols are labeled). The work gives a formal proof that for a given sized DFA the machine is identifiable with strings annotated with the NSP labels. The work also gives a L* variant (classic algorithm for identifying automata) algorithm for identifying the automaton by making use of a learned membership query and a simulated next symbol prediction, both using a trained language model (as in the original L* but with an oracle). A PAC guarantee is given that is connected to the error of the language model. This allows for extraction of DFAs from trained models, and can serve as a way to understand what “automaton” the trained model is simulating given the training data. Such an empirical study is conducted using 11 regular languages. There is also a hardness result showing how membership alone does not suffice to learn.

**Strengths:**

The work is thorough and shows us how we can cleverly extract automata from language models trained on synthetic data. This is backed up by theoretical results. The problem is well framed, the L* algorithm is nice and the guarantee accompanying it. Evaluation is done on 11 varied languages and the bridge between formal languages and neural languages is welcome.

**Weaknesses:**

The work is quite dense, and comes with a thorough appendix. I feel like it could almost have been more than a single paper.

It would have been nice to see more variations in the trained language models (architectures, hyperparams) , as an empirical study of learnability, but again that could be another paper. And same for some public LLMs.

Comparison to other ways to identify the machines could maybe have been done. I am not sure what methods would make most sense but i imagine there are some.

Tiny nitpicks: it wasn't immediately clear to me from e.g. figure 2 on its own what was being plotted until after reading the work. Maybe say (time to run L*), (number of states in found machine). Also in figure 1, the labels for the symbols make sense but they could be somehow marked as such.

**Questions:**

Did you try at all to extract automata from public LLMs for simple languages to see how they implement something like parity? E.g. to benchmark them on whether they find minimal solutions or something of that sort. I see you point out this might be overly complex machines in the appendix, but I’m curious.

Do you see a path forward to extending the work to WFSAs?

---

> ### Author Response · Authors · 2025-11-21
>
> Thank you for your thoughtful comments and time.
>
> > It would have been nice to see more variations in the trained language models (architectures, hyperparams) …
>
> We agree that a lot of things could be explored further using the algorithm. The focus of this work was to introduce the problem, theoretically characterize learnability in this new setting, and test the efficacy of the algorithm via experiments. Our current experiments focus on testing the efficacy of the algorithm itself, which is a natural step before exploring its application in other domains. We are interested in exploring its application in other domains in the near future.
>
> > (Q1) Did you try at all to extract automata from public LLMs for simple languages to see how they implement something like parity? E.g. to benchmark them on whether they find minimal solutions or something of that sort. I see you point out this might be overly complex machines in the appendix, but I’m curious.
>
> We did not try out LLMs earlier primarily due to computational challenges (discussed in the paper) with larger vocab size and model size (MQ calls), as well as some challenges around tokenization. We are currently setting up experiments and trying out a few ideas. We are hoping to have some preliminary results before the discussion period ends.
>
>
> > (Q2) Do you see a path forward to extending the work to WFSAs?
>
> We think that would be an interesting direction for future work. The problem formulation at the moment focuses on identifying the support, so it is unclear if WFSAs fit naturally here. That said, we think the current algorithm could be extended with a bit of work to learn Probabilistic DFAs by making use of next symbol probabilities instead of 0/1 labels, but it would be more interesting to analyze the learnability of more general classes of weighted automata.

---

### Author Response · Authors · 2025-11-21
**General Response**

We thank all the reviewers for their thoughtful feedback and their time. We are encouraged to see that they found our contributions to be clear and important (Rev 4bT5) and the work thorough (Rev 3o3h). In this work, we introduce a new NSP-based automata-learning framework, and we are pleased that reviewers regarded our problem formulation and setting as natural and relevant for modern language models (Rev 3o3h, VJGS, xFFN, 4bT5). We are further pleased that they found the theoretical results thoroughly backed and well-explained (Rev 4bT5), highlighted the Lemma 5.1 reduction as neat (Rev xFFN), and described the formalization of NSP queries as clean (Rev VJGS). Lastly, we are encouraged that they found our experiments extensive and well-justified (Rev 4bT5), careful and complete (Rev xFFN), and noted that NSP queries help learn faster empirically (Rev VJGS).


In this work, we formalize automata learning in a new setting tailored to language models, and theoretically characterize various aspects of learnability, such as identifiability, hardness of learning from examples, and efficient learning with generative and membership queries. We also systematically evaluate our extension of the $\mathrm{L}^{\star}$ algorithm with Transformer language models as teachers to validate its efficacy.

We have addressed the weaknesses and specific questions from each reviewer in the individual responses. Below, we summarize the key aspects of our responses.


Rev 3o3h had a few clarifying questions, and we have answered them.

Rev VJGS primarily cited the need for experiments with LLMs. We mention and expand on certain computational challenges that are discussed in the paper, and also discuss other challenges that prevent one from applying the algorithm to LLMs in a plug-and-play manner. We are also actively trying out ideas to conduct experiments with LLMs, and we are hoping to provide some results before the end of the discussion period.

Rev xFFN mentions a few weaknesses, which we already highlighted in the paper. We explain why we believe they are reasonable and do not undermine the core contributions of the paper. They also had a few questions, which we have answered.

Rev 4bT5 asked several clarifying questions, and we have provided answers to them.

---

> ### Author Response · Authors · 2025-11-25
> **Changes in the revised version**
>
> We have updated the paper with new experiments and discussion points based on the reviewers’ comments.
>
> **Experiments with LLMs.** The main addition is a section on some experiments with LLMs (Sec. I). We prompt and evaluate the ability of 1B and 7B byte-level LLMs to generate strings from 5 different regular languages. The experiments and results are divided into three parts. (i) First, we check if the DFA extracted from the LLMs with our algorithm is faithful in the sense that it has good NSP-accuracy on unseen examples. (ii) Secondly, we evaluate the ability of the extracted DFA to find erroneous strings, and (iii) lastly, we attempt to identify systematic differences in the support of the 1B and 7B LLM based on the extracted DFAs.
>
>
>
> **Additional discussion points.** We have added a couple of discussion points in Sec. H.3 (practical notes on the algorithm).
>
> **(i)** The first point discusses the effect of the tokenizer: we clarify that the algorithm always learns the token-level support of a language model. For character-level or word-level LMs, the vocabulary closely aligns with the underlying notion of a string, so the algorithm can be applied and interpreted in a straightforward way. For models with subword or BPE tokenizers, the algorithm can still be applied as is, but the extracted DFA should be interpreted as describing token-level behavior, since the mapping from token sequences to character strings is not one-to-one. See Sec. H.3 or our discussion with Reviewer VJGS for more details.
>
> **(ii)** Second, we discuss some ways to improve efficiency over a naive implementation of the algorithm by making use of an LRU cache for MQ calls and parallelizing the MQ calls in the closure step. The LRU cache is already part of our implementation, and we have also uploaded our code with a pointer to it.

---

### Meta-Review · Area_Chair_aWgQ · 2025-12-29

**Summary:**

This paper presents a novel framework for extracting formal description of neural language models in the form of deterministic finite automata. Their framework considers a Next Symbol Prediction (NSP) setting where the learner receives for each prefix whether the prefix is in the language or which next symbol can lead to an accepting string. The authors show that the class of DFA with at most $n$ states is identifiable from positive examples augmented with these NSP labels but remains computationally hard in the PAC-setting and exact identification cannot be achieved in polynomial time using membership queries. They propose an extension that is able to efficiently PAC-learn DFA using a language model-based teacher able to answer membership queries and generating positive strings conditioned on prefix prompts by means of an extension of the L* algorithm. The paper provides theoretical results and en experimental evaluation on Transformer-based language models.


Based on the released reviews, the following strengths and weaknesses have been identified:
-Reviewer 3o3h mentions that the work is thorough, the problem is well framed, the solution is nice, justified by theoretical results with interesting empirical evaluation.
On the hand, the work is dense. The experimental evaluation could have more variations in trained models and other ways could have been considered for the comparisons.
-Reviewer VJGS indicated that the formalization with NSP queries is clean, the problem worth to be studied and the result is interesting. The paper is formal and self-contained.
On the other hand, the experimental evaluation is mainly based on small transformers on toy languages and more general LLM could have been considered. Additionally, the fact that LLMs consider of tokens and not characters should lead to a discussion.
-Reviewer xFFN highlights that NSP exactly captures what a practitioner can extract from a language model, the focus on support is natural and practically important. The justification of the extension of L* also is neat. The evaluations are careful and complete.
On the other hand, the paper appears limited for dealing with large vocabularies (but is honest on this). The PAC-guarantee is restricted to the language model distribution. The model might be negatively impacted by long or non regular support.
-Reviewer 4bT5 identifies that the paper is well structured and written. The contributions are clear and important, the proposed setting improves existing ones to a more useful one with modern language models. The theoretical results are well supported and explained. The empirical evaluation is sound. There is an interesting FAQ section. The reviewer does not identify real weaknesses, except that the algorithm cannot be applied to actual LLMs.

Overall, the trend for this paper is very positive. All reviewers have underlined that the paper is thorough, the contributions are nice and well justified, with a novel setting that is well suited to actual language models.
On the hand hand, some limitations have raised. Authors have tried to answer all the issues and questions asked by the reviewers. In their revision they add an additional experiment with 2 LLMs and add discussions on the limitation due to the effect of the tokenizer and how to improve the efficiency of the implementation.

**Reviewer Concerns:**

For Reviewer 3o3h, authors have answered the different questions during rebuttal. Authors did not address the idea of providing more variations in the experimental evaluation, but the evaluation was already good and the reviewer mentions that it could be another paper.

For reviewer VJGS, authors have provided answers and discussions on the addition of experiments with LLMs. They provided an additional experimental study with 2 LLMs and discuss the challenge with the tokenizer. The main issues of the reviewer were addressed and acknowledged that he is on the same line with the authors.

For reviewer xFFN, authors have provided in the rebuttal some explanations on why the weaknesses raised are reasonable and they have provided answers to the few questions of the reviewers. In the revision, a special note on the efficiency of the algorithm has been added.

For reviewer 4bT5, authors have provided answers to the clarifying questions of the reviewer. The reviewer answered that the answers addressed his questions very well.

**Reviewer Scores:**

R 3o3h proposed an 8. His weaknesses were not that strong and reviewers provided some answers. He is likely to maintain his evaluation.

R VJGS proposed a 6. He was happy with the answers. He supported the paper and mention that he would update his score.

R xFFN proposed a 6. Authors provided comment to the weaknesses justifying that they are not too strong and add a section on the implementation. Not sure that the reviewer would have improved his score, but his strengths were rather positive on the work, so he would at least have kept his score.

R 4bT5 proposed 8 answered that he maintained his positive evaluation on the work and that he hopes the paper will be accepted.

---

### Decision · Program_Chairs · 2026-01-26

Accept (Poster)